# Conformational heterogeneity of the BTK PHTH domain drives multiple regulatory states

**David Yin-wei Lin**[1]**, Lauren E Kueffer**[1]**, Puneet Juneja**[2]**, Thomas E Wales**[3]**, John R Engen**[3]**, Amy H Andreotti**[1]*****

[1]Roy J. Carver Department of Biochemistry, Biophysics and Molecular Biology, Iowa State University, Ames, United States; [2]Cryo-EM Facility, Office of Biotechnology, Iowa State University, Ames, United States; [3]Department of Chemistry and Chemical Biology, Northeastern University, Boston, United States

**\*For correspondence:**
amyand@iastate.edu

**Abstract** Full-length Bruton's tyrosine kinase (BTK) has been refractory to structural analysis. The nearest full-length structure of BTK to date consists of the autoinhibited SH3–SH2–kinase core. Precisely how the BTK N-terminal domains (the Pleckstrin homology/Tec homology [PHTH] domain and proline-rich regions [PRR] contain linker) contribute to BTK regulation remains unclear. We have produced crystals of full-length BTK for the first time but despite efforts to stabilize the autoinhibited state, the diffraction data still reveal only the SH3–SH2–kinase core with no electron density visible for the PHTH–PRR segment. Cryo-electron microscopy (cryoEM) data of full-length BTK, on the other hand, provide the first view of the PHTH domain within full-length BTK. CryoEM reconstructions support conformational heterogeneity in the PHTH–PRR region wherein the globular PHTH domain adopts a range of states arrayed around the autoinhibited SH3–SH2–kinase core. On the way to activation, disassembly of the SH3–SH2–kinase core opens a new autoinhibitory site on the kinase domain for PHTH domain binding that is ultimately released upon interaction of PHTH with phosphatidylinositol (3,4,5)-trisphosphate. Membrane-induced dimerization activates BTK and we present here a crystal structure of an activation loop swapped BTK kinase domain dimer that likely represents the conformational state leading to trans-autophosphorylation. Together, these data provide the first structural elucidation of full-length BTK and allow a deeper understanding of allosteric control over the BTK kinase domain during distinct stages of activation.

## eLife assessment

BTK, a TEC-family tyrosine kinase activated by the B-cell antigen receptor, contains a variety of regulatory domains and it is subject to complex regulation by membrane phospholipids, protein ligands, phosphorylation, and dimerization. This study presents **convincing** evidence, utilizing various biophysical techniques, to support a model for BTK activation that will be **valuable** for the field. Overall, the study enhances the understanding of BTK's activation mechanism, autoinhibition, and allosteric control, challenging previous assumptions about BTK.

## Introduction

The TEC family kinase, BTK (Bruton's tyrosine kinase) is best known as the target of ibrutinib (IMBRU-VICA), the first-in-class covalent kinase active site inhibitor used to treat chronic lymphocytic leukemia, mantle cell lymphoma, Waldenström's macroglobulinemia, and chronic graft-versus-host disease. The TEC kinases are the second largest subfamily of non-receptor tyrosine kinases in the human genome

**Figure 1.** Current Bruton's tyrosine kinase (BTK) structural data. (**a**) Comparison of the SRC and TEC domain architectures. Linker regions and domains are labeled, residue numbering shows BTK domain boundaries. The 'Src module' is the SH3–SH2–kinase region shred by both families. (**b**) Autoinhibited BTK core (Src module). The compact structure of the SH3–SH2–kinase region of BTK is exacted from the domain swapped dimer structure (PDB: 4XI2) solved by **Wang et al., 2015**. The three domains (SH3, SH2, and kinase), the SH2–kinase linker, the activation loop and the active site are labeled. (**c**) Structure of the BTK Pleckstrin homology/Tec homology (PHTH) domain bound to inositol 1,3,4,5-tetrakisphosphate (IP$_4$; PDB: 1B55) (**Baraldi et al., 1999**). The monomer is shown for clarity and the TH region bound to Zn$^{2+}$ is circled. (**d**) Structure of the tethered PHTH–kinase construct (PDB: 4Y93). (**e**) Solution-based mapping of BTK PHTH interaction across the activation loop face of the kinase domain.

after the SRC family (**Yu and Smith, 2011**; **Mano, 1999**; **Smith et al., 2001**). The TEC and SRC kinases share the SH3–SH2–kinase domain arrangement (the 'Src module') (**Figure 1a**). Structures of the auto-inhibited SRC family kinases were solved in 1997 revealing, for the first time, the compact arrangement of the SH3 and SH2 domains assembled onto the distal side of the catalytic kinase domain (**Xu et al., 1997**). Despite the shared 'Src module' of the SRC and TEC families, 18 years passed before

an autoinhibited structure of one of the TEC kinases was resolved by X-ray crystallography. In 2015, the BTK SH3–SH2–kinase fragment was crystallized (*Wang et al., 2015*). This BTK fragment crystallizes as a domain swapped dimer; the SH2 domain opens to allow the B helix of one chain to pack against the β-sheet of the other chain. One-half of the domain swapped BTK structure represents the current model for autoinhibited BTK and closely resembles the compact structure of autoinhibited SRC kinases (*Figure 1b*).

Outside of the Src module, the sequence and domain structures of the TEC and SRC kinases diverge. The SRC kinases contain a C-terminal tail (absent in the TEC family) that, upon phosphorylation on a conserved tyrosine, binds to the SH2 domain in an intramolecular fashion to stabilize the autoinhibited Src module. BTK instead stabilizes the SH2 autoinhibitory pose via a conserved acidic side chain at the end of the kinase domain (*Joseph et al., 2017*). The SRC kinases also contain an N-terminal SH4-unique domain region (*Figure 1a*) that is intrinsically disordered and has not been resolved in the available crystal structures. The SRC kinases are myristoylated or palmitoylated in the SH4 domain driving membrane association (*Koegl et al., 1994*). In contrast, BTK contains an extended N-terminal region (PHTH–PRR) consisting of the phospholipid-binding Pleckstrin homology (PH) domain followed by a Tec homology (TH) domain and a long linker housing two proline-rich regions (PRR) (*Figure 1a*). The folded N-terminal PH domain of BTK has been crystallized (*Baraldi et al., 1999*) revealing a binding site for inositol 1,3,4,5-tetrakisphosphate ($IP_4$), the headgroup of its target phosphatidylinositol (3,4,5)-trisphosphate ($PIP_3$) ligand (*Figure 1c*). The crystal structure shows the TH domain bound to zinc and closely associated with the PH domain; the folded N-terminal domain of BTK is hereafter referred to as PHTH. The BTK PHTH domain crystallizes as a homodimer, named for the preeminent structural biologist, Matti Saraste, who solved the first BTK PHTH structure (*Hyvönen and Saraste, 1997*).

The N-terminal PHTH–PRR domains account for over 30% of the BTK sequence. The BTK PHTH domain structure has been solved both as the isolated domain and in a construct where PHTH is covalently attached to the BTK kinase domain through a short tether (*Wang et al., 2015*). The tightly tethered structure revealed an interface between the PHTH domain and the N-lobe of the kinase domain (*Figure 1d*) but the contacts are mutually exclusive with the contacts between the BTK SH3 and kinase domain in the autoinhibited 'Src module' (*Figure 1b*) raising questions about whether both the SH3 and PHTH domain associate with the kinase domain simultaneously. At the same time, we conducted solution-based Nuclear Magnetic Resonance and hydrogen/deuterium exchange mass spectrometry (HDXMS) studies (*Devkota et al., 2017*; *Amatya et al., 2019*) that point to autoinhibitory interactions between the PHTH domain and the activation loop face of the BTK kinase domain (*Figure 1e*). In this case, the mapped interface reflects intermolecular contacts that result upon addition of excess PHTH domain to the isolated BTK kinase domain (NMR) or comparison of full-length BTK with the SH3–SH2–kinase fragment (HDXMS). Thus, the BTK structural data reported to date support two poses for the N-terminal PHTH domain: PHTH domain landing on the distal face of the kinase domain N-lobe (*Figure 1d*) and PHTH domain binding to the activation loop face of the kinase domain (*Figure 1e*).

Here, we set out to solve the structure of the full-length BTK protein. Our crystallization target included the N-terminal PHTH domain and the long PRR linker region followed by the SH3, SH2, SH2–kinase linker, and kinase domains (*Figure 1a*). Mutations were introduced into the SH3–SH2–kinase core in an effort to stabilize the compact autoinhibited conformation. We reasoned that this approach might encourage the PHTH domain to adopt a stable autoinhibited state or states. Despite efforts to optimize the autoinhibitory conformation of BTK, the crystallography presented here reveals the same domain swapped dimer solved previously and no electron density is observed for the PHTH-linker residues. Further analyses using both small angle X-ray scattering (SAXS) and cryo-electron microscopy (cryoEM) support a BTK autoinhibited state wherein the core SH3–SH2–kinase region adopts the compact monomeric form much like that observed in the early SRC kinase structures. CryoEM allows the PHTH domain to be directly observed; the globular PHTH adopts a range of conformational states that surround but do not contact the core 'Src module'. To revisit the PHTH domain crystallographically, we lengthened and redesigned the tether between PHTH and kinase domains to provide greater conformational freedom between domains. The resulting crystal structure revealed yet another binding pose for the BTK PHTH domain on the BTK kinase domain and mutational analysis supports a regulatory role for this interface in the full-length kinase. Finally, we have also captured a structure of a BTK kinase domain dimer revealing a potential arrangement of this domain

undergoing trans-autophosphorylation following PIP$_3$-mediated dimerization. Together, the structural and biochemical data presented here provide new insights into the allosteric control of BTK catalytic function. These findings create new opportunities to target this kinase in the context of disease and/or drug resistance.

## Results

### Construct design and optimization

To stabilize the autoinhibited conformation of full-length BTK for crystallography, we first optimized the inhibitory contacts between the SH3 domain and the N-lobe of the BTK kinase domain. Following previous efforts that successfully stabilized this region in a related SRC family kinase (*Lerner et al., 2005*), we substituted key residues within the SH2-linker sequence with proline (construct '4P' in *Figure 2a*). The catalytic activity of the full-length BTK 4P variant, measured by autophosphorylation on Y511 in the BTK activation loop, is reduced compared to wild-type BTK (*Figure 2b*). In addition, the melting temperature ($T_m$) of BTK 4P is 4°C higher than wild-type BTK consistent with stabilization of the overall fold (*Figure 2c*, *Figure 2—figure supplement 1*). Next, we tested the effect of mutation of L390 to phenylalanine on the activity and stability of full-length BTK. L390 resides in the SH2-linker and has been shown to stabilize the hydrophobic stack (W421 and Y461) in the kinase domain N-lobe (*von Raußendorf et al., 2017*). The aromatic phenylalanine is found in other members of the TEC family suggesting it is well tolerated in this position. We find the L390F mutation does not change the activity of full-length BTK but it does increase the $T_m$ by 1.5°C compared to wild-type BTK (*Figure 2b , c* and *Figure 2—figure supplement 1*). We therefore combined the 4P and L390F mutations (BTK 4P1F) and observe a further decrease in autophosphorylation activity and 4.5°C increase in $T_m$ (*Figure 2b, c* and *Figure 2—figure supplement 1*). Lastly, we compared the BTK 4P1F mutant to full-length BTK containing ITK activation loop residues (L542M S543T, V555T, R562K, S564A, and P565S) instead of the wild-type BTK sequence. We have previously show that incorporation of these ITK activation loop residues results in dampened loop dynamics that favor the autoinhibited conformation (*Joseph et al., 2013*). As well, previous BTK crystallography efforts showed that these changes in the activation loop are required for crystallization (*Wang et al., 2015*). As expected, the activity of full-length BTK with the ITK loop substitution is low and we find that this sequence change also stabilizes the overall protein as the $T_m$ is 2.5°C higher than wild-type BTK (*Figure 2b, c* and *Figure 2—figure supplement 1*). These data suggest that targeted modifications within the SH2-linker region and activation loop of BTK result in stabilization of the autoinhibited conformation.

Previous work on BTK made use of SAXS analysis to characterize both the full-length protein and domain fragments (*Duarte et al., 2020*; *Márquez et al., 2003*). Here, we used SAXS to further characterize the effect of the sequence changes introduced to stabilize the autoinhibited state. Consistent with the previous observations (*Duarte et al., 2020*; *Márquez et al., 2003*), the SH3–SH2–kinase fragment adopts a compact conformation while the full-length BTK protein is more extended (*Figure 2d, e*). The SAXS-derived structural parameters ($R_g$ and $D_{max}$) suggest that, compared to wild-type, the 4P1F BTK mutations have no effect on the average conformation of the SH3–SH2–kinase fragment. Ab initio shape reconstructions superimpose with the autoinhibited SH3–Sh2–kinase model derived from the previously solved structure (PDB: 4XI2) (*Figure 2e*, top). In contrast, the volumes of both the ab initio envelopes of full-length wild-type and 4P1F BTK mutant are elongated (*Figure 2e*, bottom). The full-length 4P1F mutant exhibits a somewhat more compact envelope (decreased particle dimension compared to wild-type full-length BTK) possibly consistent with the stabilized SH3–SH2–kinase core shifting the conformational preferences of the N-terminal PHTH–PRR region (*Figure 2d, e*). Dimensionless Kratky and Grunier plots of the recombinant BTK proteins are provided in *Figure 2—figure supplement 1*.

With the stabilized full-length BTK mutant showing low activity (*Figure 2b*) and a more compact conformational ensemble in solution (*Figure 2d, e*), we built a full-length BTK construct for crystallization screens that incorporated the 4P1F mutations and the ITK loop (*Figure 3a*). In addition, the catalytic K430 was mutated to arginine to eliminate kinase activity that could produce heterogeneity. This construct produced very small crystals in multiple crystallization conditions. To improve the size and quality of the crystals we introduced mutations at three side chains in the SH2 domain predicted to have high surface entropy (*Goldschmidt et al., 2007*). The resulting construct, referred

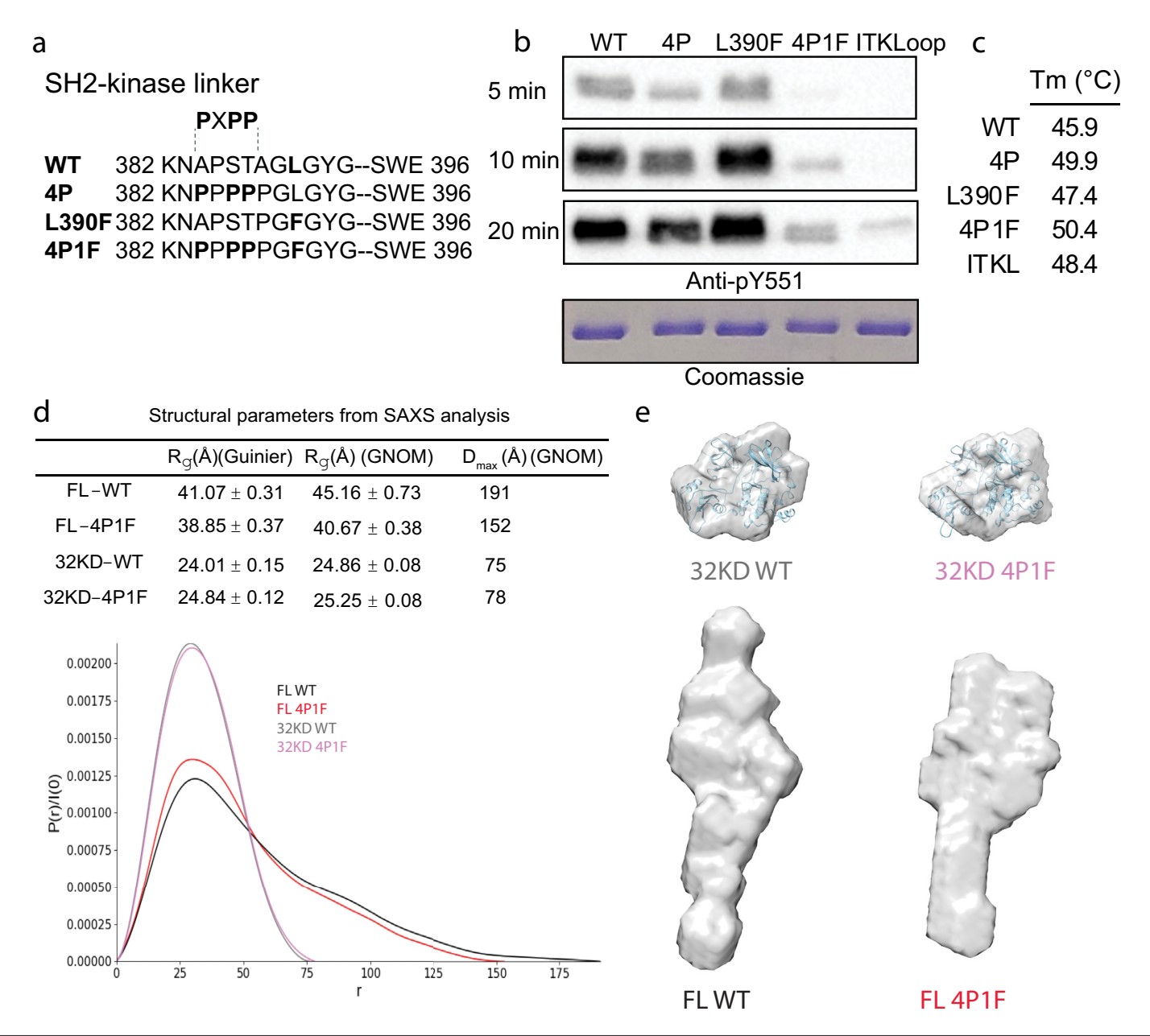

**Figure 2.** Stabilization of the Bruton's tyrosine kinase (BTK) SH3–SH2–kinase core. (**a**) Mutations introduced into the SH2–kinase linker region of BTK (residues 382–396). PXPP indicates the region that adopts the left-handed, type II polyproline helix in the autoinhibited structure of BTK SH3–SH2–kinase. (**b**) Western blot showing the kinase activity of wild-type (WT) BTK, 4P, L390F, 4P1F, and ITKLoop BTK variants. Autophosphorylation on BTK is monitored using an anti-pY551 antibody and total protein levels are monitored by Coomassie stain. (**c**) Melting temperatures of BTK WT and variants (see *Figure 2—figure supplement 1*). (**d**) Distance distribution functions and structural parameters ($R_g$ and $D_{max}$) comparing the SH3–SH2–kinase fragment and full-length BTK (wild-type and 4P1F). (**e**) Surface representation of ab initio envelope reconstructions obtained from small angle X-ray scattering (SAXS) superimposed on the crystal structures for the BTK SH3–SH2–kinase fragment (top). Elongated envelopes for both full-length wild-type (FL WT) BTK and the full-length 4P1F mutant of BTK are shown without structure superposition (bottom). *Figure 2—figure supplement 1* provides Guinier and Kratky plots for all four BTK proteins. SASBDB accession codes are as follows: SASDRB9, SASDRC9, SASDRD9, and SASDRE9.

The online version of this article includes the following figure supplement(s) for figure 2:

**Figure supplement 1.** Tm and SAXS analyses.

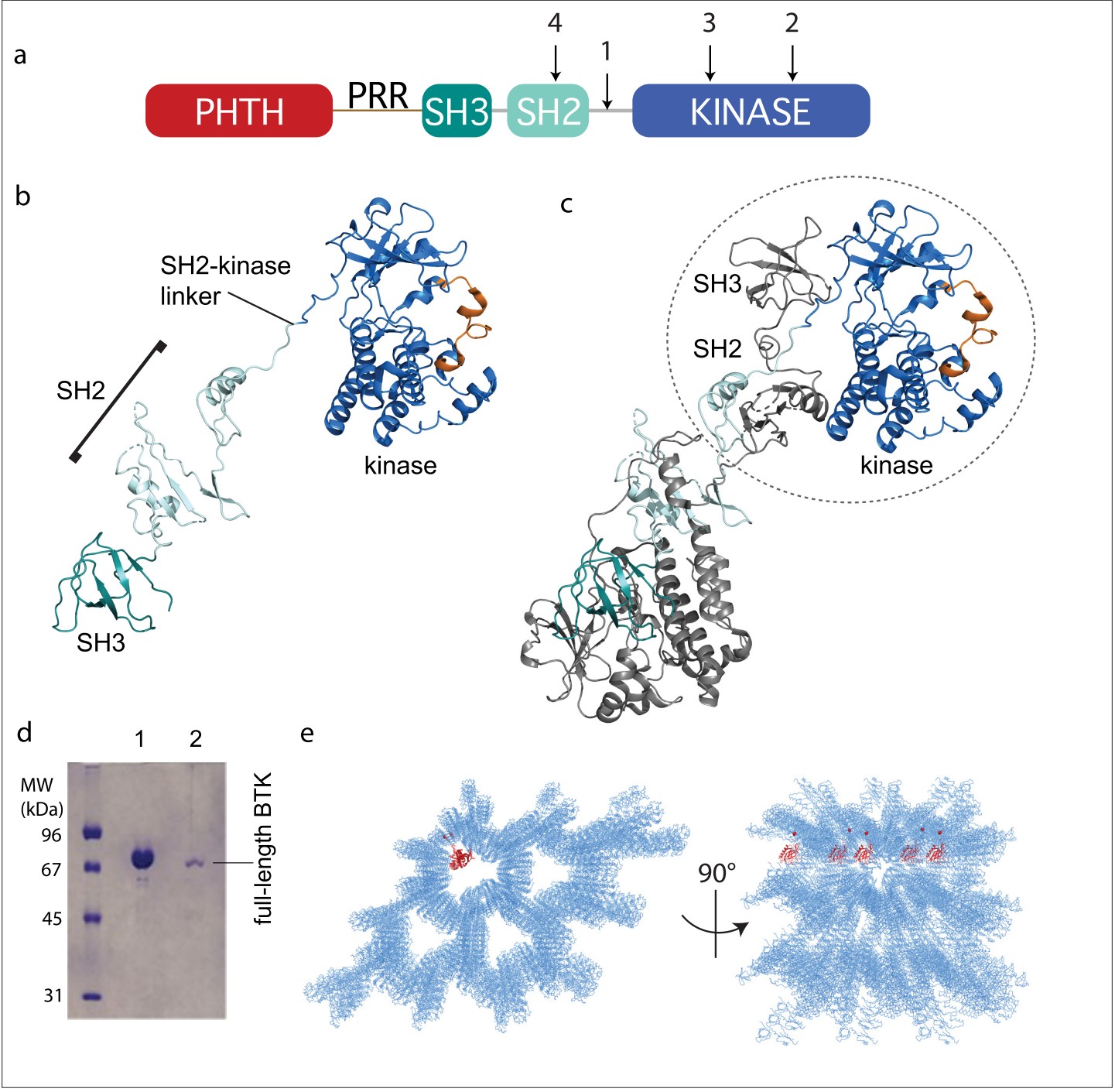

**Figure 3.** Crystallization of full-length Bruton's tyrosine kinase (BTK). (**a**) The crystallization target, full-length BTK with stabilized core (FL BTKsc), included (1) SH2–kinase linker mutations (4P1F: A384P, S386P, T387P, A388P, and L390F); (2) activation loop mutations (ITKLoop: L542M, S543T, V555T, R562K, S564A, and P565S); (3) catalytic residue mutation (K430R); and (4) surface entropy reduction mutations (E298A, K300A, and E301A). The N-terminal domains, Pleckstrin homology/Tec homology (PHTH)–proline-rich region (PRR)–SH3, are wild-type BTK sequence. (**b, c**) Structure of the BTK domain swapped dimer that results from crystallization of full-length BTK (PDB: 8GMB). PHTH–PRR region is missing from the electron density. One SH3–SH2–kinase monomer is shown in (**b**) and the autoinhibited SH3–SH2–kinase arrangement is circled in (**c**). Domain colors match those in (**a**) and the activation loop in the kinase domain is orange. (**d**) Sodium dodecyl sulfate–polyacrylamide gel electrophoresis (SDS–PAGE) showing full-length BTK protein from crystals. Lane 1 is a purified full-length BTK control and lane 2 is protein derived from harvested and washed crystals. (**e**) Two views of crystal packing with the PHTH domain (red) modeled into one of the solvent channels.

to as full-length BTK with stabilized core (FL BTKsc) (*Figure 3a*), provided diffraction quality crystals (5–7 days at 4°C) both in the apo form and in the presence of various active site inhibitors. The highest resolution data collected (3.4 Å) was in the presence of the GDC-0853 (fenebrutinib) inhibitor.

## BTK crystal structure

Despite efforts to stabilize the autoinhibited form, the electron density derived from crystals of full-length BTK (FL BTKsc) defines the same domain swapped dimer observed for the previously solved structure of the BTK SH3–SH2–kinase fragment (*Wang et al., 2015*; *Figure 3b, c*). A 16.9° rotation along residues 346–384 in the SH2 and SH2–kinase linker is observed for the domain swapped dimer structure solved here compared to that solved previously (*Wang et al., 2015*). Notably, there is no PHTH domain density visible even after the structure is fully refined. To confirm that the crystals contain the full-length BTK protein and not a degradation fragment, we harvested, washed, and analyzed the crystals by sodium dodecyl sulfate–polyacrylamide gel electrophoresis (SDS–PAGE) (*Figure 3d*). The result shows the crystal is full-length BTK protein with no evidence of smaller BTK fragments. Finally, we built five additional full-length BTK constructs with varying mutations/deletions within the long PRR containing linker between the PHTH and SH3 domains (see Materials and methods). We find that four of the five constructs crystalize in a manner identical to FL BTKsc, which contains the wild-type sequence in the PHTH–PRR region, and no electron density is observed for the PHTH domain in any of these crystals (one construct did not form crystals).

The solvent content in the crystals of full-length BTK is high (67%) and crystal packing shows large solvent channels (*Figure 3e*). Given the absence of PHTH domain electron density, we asked whether the previously solved PHTH domain structure could fit into the solvent channels. Placing the PHTH structure at the N-terminus of the BTK SH3–SH2–kinase model shows that the PHTH domain is readily accommodated within the solvent channels of the crystal (*Figure 3e*). The crystallography results are consistent with a flexible N-terminal PHTH domain with the caveat that crystal packing might limit native autoinhibitory contacts between the PHTH and SH3–SH2–kinase regions. To address this, we examined the accessibility of the previously identified PHTH domain-binding site on the activation loop face of the BTK kinase domain (*Amatya et al., 2019*) in the context of symmetry related molecules in the crystal. The C-lobe G-helix portion of the PHTH/kinase interface mapped previously by solution methods (*Amatya et al., 2019*), appears inaccessible while the activation loop itself and the portion of the N-lobe previously identified within the PHTH/kinase interface are accessible. Thus, it is possible that the PHTH domain mobility in the crystal is due to steric occlusion of the previously identified regulatory site. To further explore the domain arrangements within full-length BTK we next turned to single-particle cryoEM.

## CryoEM studies of full-length BTK

Using the same purified BTK protein used for structure determination by crystallography (FL BTKsc), we obtained cryoEM reconstructions of full-length BTK (*Figure 4*). BTK particles were nicely distributed in the raw images (*Figure 4—figure supplement 1*). After several rounds of 2D classification, the 2D classes resembling the autoinhibited SH3–SH2–kinase core start to become visible (*Figure 4a, b*, *Figure 4—figure supplement 1*). An extra unstructured density mass positioned adjacent to the main SH3–SH2–kinase core in addition to a globular density distant from the core are evident in the 2D classes and were persistently present after discarding specific 2D classes and repeated 2D classification. From ab initio reconstruction and refinement, four 3D classes were obtained (*Figure 4—figure supplement 1*); three of which (Classes 0, 1, and 3) were of sufficient quality to fit models of BTK SH3–SH2–kinase (*Figure 4c–e*). The presence of the monomeric SH3–SH2–kinase region in all of the 3D reconstructions is significant as all BTK crystal structures to date have revealed a domain swapped dimer with two polypeptide chains contributing to the autoinhibited conformation.

Two of the three 3D reconstructions (Classes 0 and 1) show density possibly representing the N-terminal PHTH domain and adjacent PRR linker region (*Figure 4c, d*). The PHTH domain structure fits into the extra globular density that sits outside of the SH3–SH2–kinase core, and in one reconstruction, continuous density is observed from the SH3–SH2–kinase core to the globular domain (*Figure 4c*). The remote globular density is not visible in Class 3, but similar to Classes 0 and 1, continuous unresolved density next to the SH3–SH2–kinase core is present possibly indicating the location of the PRR linker region that is covalently connected to the SH3 domain (*Figure 4e*).

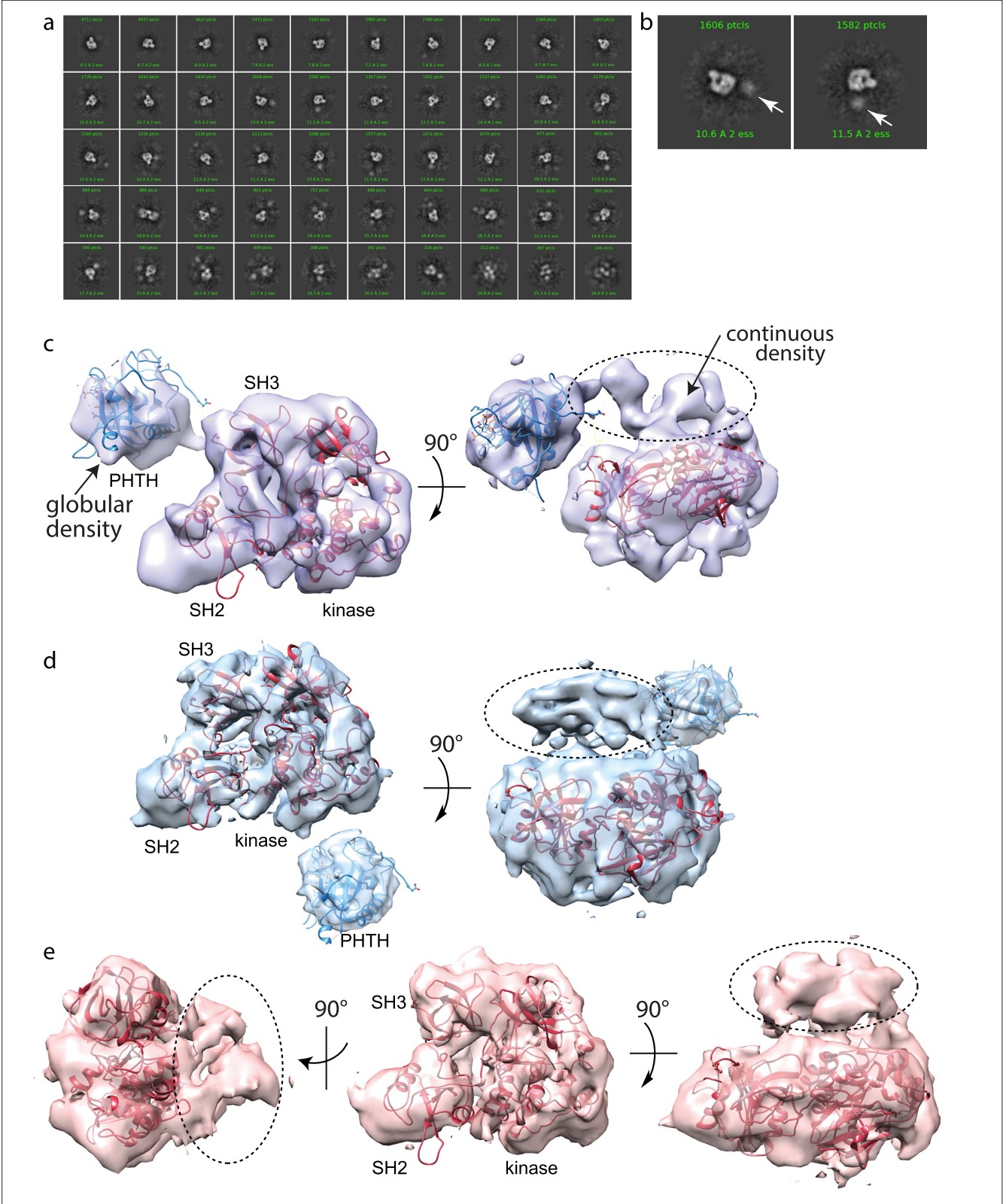

**Figure 4.** Full-length Bruton's tyrosine kinase (BTK) cryo-electron microscopy (cryoEM) analysis. (**a**) 2D class averages of full-length BTK. (**b**) Representative 2D class averages. White arrow indicates extra density adjacent to the BTK SH3–SH2–kinase core density. (**c–e**) Three final 3D reconstructions (see workflow in *Figure 4—figure supplement 1*). (**c**) Two views of the Class 0 map with the SH3–SH2–kinase fragment (PDB: 8GMB) fit into the larger density and monomeric Pleckstrin homology/Tec homology (PHTH) domain (PDB: 1B55) fit into the smaller globular density. Continuous

*Figure 4 continued on next page*

*Figure 4 continued*

density between the large and small density is indicated with a dashed circle. (**d**) Two views of the Class 1 map with globular density fit as described for (**c**). The smaller globular density is located in a distinct position with respect to the SH3–SH2–kinase core compared to that shown in (**c**). Additionally, unmodeled density is observed adjacent to the BTK SH3–SH2–kinase core (dashed circle); the position is similar to the continuous density observed in Class 0. (**e**) Three views of the Class 3 map with fitted BTK SH3–SH2–kinase core and unmodeled density that is in the same location as that in (**d**). EMDB accession codes are as follows: EMD-40585, EMD-40586, and EMD-40587. Map fitting without user input was also carried out using Situs (**Wriggers, 2012**). The result of that fitting is comparable to results obtained using Chimera.

The online version of this article includes the following figure supplement(s) for figure 4:

**Figure supplement 1.** Workflow showing cryo-electron microscopy (cryoEM) analysis of full-length Bruton's tyrosine kinase (BTK).

**Figure supplement 2.** Deuterium uptake curves for the Pleckstrin homology/Tec homology (PHTH) through SH3 domain of Bruton's tyrosine kinase (BTK).

HDXMS data on wild-type, full-length BTK (***Figure 4—figure supplement 2***) are consistent with an exposed PRR linker region surrounded by two globular folded domains (PHTH and SH3). Deuterium uptake reaches its maximum at the earliest time points for peptides spanning the linker between PHTH and SH3 consistent with high solvent accessibility and the lack of a well-defined hydrogen-bonding network. In contrast, the deuterium uptake curves for peptides derived from all other regions of BTK (PHTH, SH3–SH2, kinase domains) are consistent with folded domains.

## Pursuing the N-terminal PHTH domain

Results from crystallography and cryoEM presented here suggest the PHTH domain is not stably associated with the autoinhibited SH3–SH2–kinase core of BTK but instead samples a broad conformational ensemble within the context of full-length BTK that may include transient interaction with the autoinhibited core. Indeed, we and others have previously implicated the PHTH domain in autoinhibitory contacts with the kinase domain of BTK (***Wang et al., 2015***; ***Devkota et al., 2017***; ***Amatya et al., 2019***). Specifically, we have mapped an interaction between the PHTH domain and the activation loop face of the BTK kinase domain (***Figure 1e***) using solution methods. As well, the BTK fragment lacking the PHTH–PRR region is more active than the full-length protein (***Wang et al., 2015***) and the same work suggested that mutation of specific surface exposed side chains within the PHTH domain (R133 and Y134) results in modest activation of BTK. The latter was based on the tightly tethered PHTH–kinase crystal structure (***Figure 1d***, PDB: 4Y93) (***Wang et al., 2015***). The location of the PHTH domain in the tightly tethered structure (PDB: 4Y93) is sterically incompatible with the autoinhibited SH3–SH2–kinase structure, and so the authors used molecular dynamics simulations to arrive at a model for full-length autoinhibited BTK that accommodates both the SH3 and PHTH domain on the kinase N-lobe (***Wang et al., 2015***). Given the short linker used in that work, the steric clash between SH3 and PHTH domains in the two separate structures, and the fact that our X-ray and cryoEM analyses of full-length BTK do not reveal precise PHTH autoinhibitory contacts, we decided to pursue the role of PHTH in BTK autoinhibition further by crystallizing a redesigned tethered PHTH–kinase construct.

The first step in our redesign was to lengthen the linker between PHTH and kinase domain to reflect the flexibility and distance between the N- and C-terminal domains of BTK. We inserted GGSGG repeats that were either 22 or 32 amino acids in length (***Figure 5a***). This linker entirely replaced the PRR, SH3, SH2, and SH2-linker sequence. We intentionally excluded the SH2–kinase linker from the PHTH–kinase construct since this linker, in particular L390, mediates autoinhibitory contacts between the SH3 domain and the kinase domain N-lobe. In addition, the new [1]PHTH[171]-G(GGSGG)$_{4/6}$G-[396]kinase[659] constructs included PHTH β5–β6 loop mutations (Q91A, I92A, I94A, and I95A) to prevent formation of the 'Saraste dimer' since that structure is associated with BTK activation (***Hyvönen and Saraste, 1997***). As before, the K430R and ITK activation loop mutations were incorporated to facilitate crystallization. The GDC-0853 inhibitor was included in all crystallization conditions. Both redesigned constructs yielded crystals in the same crystallization conditions (see Materials and methods), but crystals from the construct containing the 22 amino acid G(GGSGG)$_4$G linker yielded larger and higher quality diffracting crystals compared to the construct containing the longer 32 amino acid linker.

The structure of the PHTH–G(GGSGG)$_4$G–kinase bound to GDC-0853 was determined to 2.1 Å (***Figure 5b***). The kinase domain adopts the inactive conformation similar to other structures of inactive BTK (Root Mean Square Deviation of 0.43 Å relative to the structure of the isolated inactive kinase

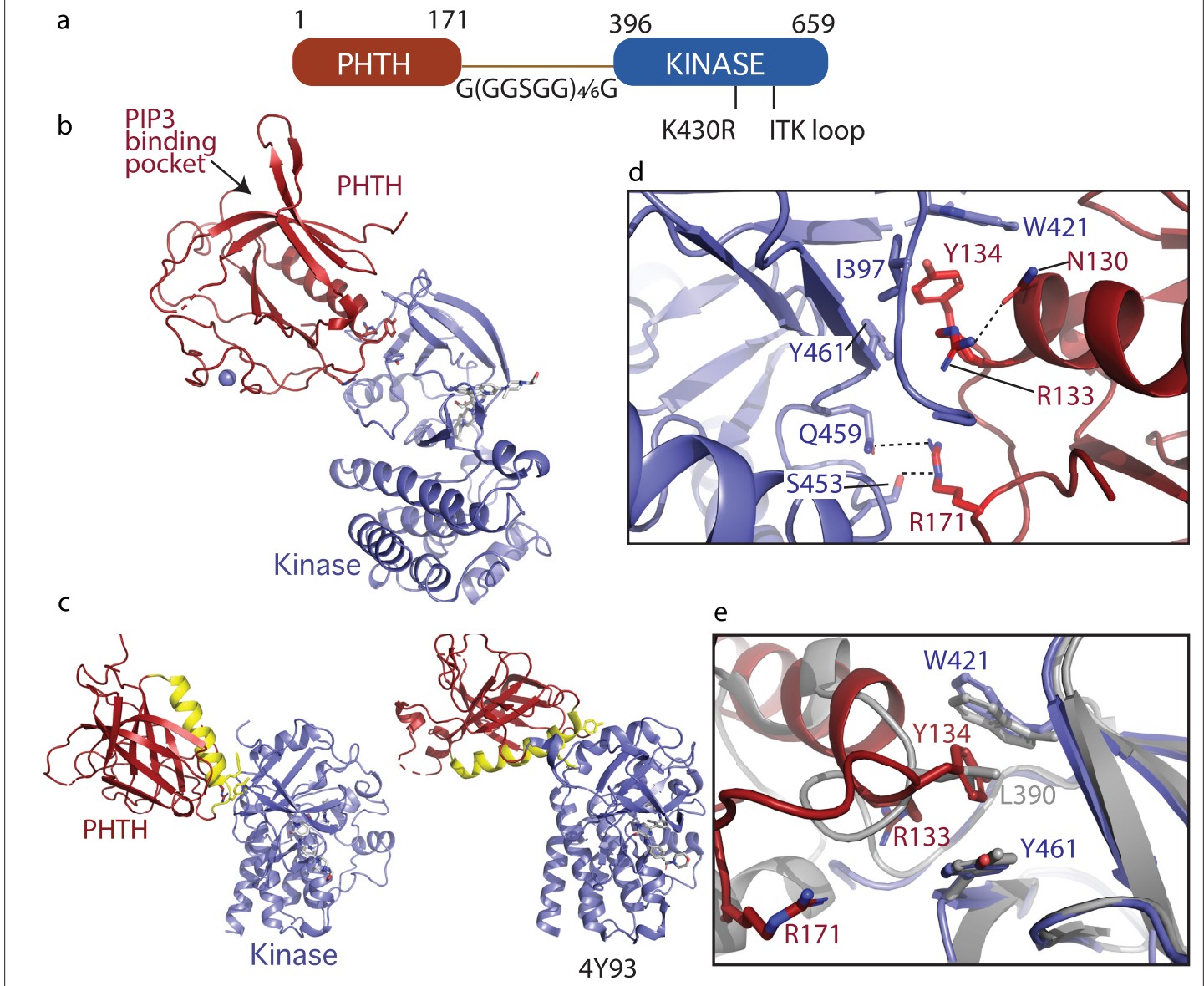

**Figure 5.** Crystallization of loosely tethered Pleckstrin homology/Tec homology (PHTH)–kinase. (**a**) Architecture of the loosely tethered PHTH–kinase constructs used for crystallography. (**b**) Crystal structure of the PHTH–kinase protein (PDB: 8S93). The PHTH domain (red) docks onto the back of the kinase domain N-lobe (blue). The location of the phosphatidylinositol (3,4,5)-trisphosphate (PIP$_3$)-binding pocket on PHTH is indicated. (**c**) Direct comparison of the loosely tethered PHTH–kinase structure solved here (left, PDB: 8S93) and the more tightly tethered PHTH–kinase structure solved previously (right, PDB: 4Y93, right). The PHTH domain helix is colored yellow and the kinase domains are in the same orientation to emphasize the difference between the PHTH domains in the two structures. (**d**) PHTH/kinase interface. PHTH side chains R133, Y134, and R171 (red) make contacts to the Bruton's tyrosine kinase (BTK) domain (blue). Dotted lines indicate hydrogen bonds. (**e**) Close-up view of the hydrophobic stack (flanked by W421 and Y461) on the kinase domain N-lobe. The PHTH Y134 residue (red) inserts into the hydrophobic stack (blue) in the loosely tethered PHTH–kinase structure solved here while L390 from the SH2–kinase linker (gray) completes the hydrophobic stack in the previously solved PHTH–kinase structure (PDB: 4Y93).

domain [PDB: 5VFI] and 0.93 Å relative to the structure of autoinhibited SH3–SH2–kinase; *Figure 3c*). The PIP$_3$-binding pocket is sterically accessible (*Figure 5b*) as was observed in the tightly tethered BTK PHTH/kinase complex. Otherwise, the orientation and specific contacts between the PHTH and kinase domains in this new, loosely tethered complex (PHTH–G(GGSGG)$_4$G–kinase) are quite distinct from the previously solved structure of the PHTH domain tethered to the kinase domain (PDB: 4Y93) (*Figure 5c*). In the structure solved here (*Figure 5c*, left), the longer linker between domains allows the PHTH domain to bind to the β-strands of the kinase domain N-lobe instead of the C-helix-binding

site defined previously (*Wang et al., 2015*). Three PHTH domain side chains, R133, Y134, and R171, provide key interactions with the kinase domain in this new structure (*Figure 5d*). The Y134 side chain inserts into the hydrophobic stack (*von Raußendorf et al., 2017*; *Gonfloni et al., 1997*) created by W421 and Y461 on the kinase domain N-lobe in a manner that mimics the L390 side chain position in the autoinhibited SH3–SH2–kinase structure (*Figure 5d, e*). The guanidinium group of R171 participates in hydrogen bonds with the side chains of S453 and Q459 in the kinase domain (*Figure 5d*). The guanidinium group of R133, in contrast, points back into the PHTH domain forming a hydrogen bond with the PHTH N130 side chain (*Figure 5d*) while the hydrophobic methylene groups of the R133 side chain contact I397 and Y461. The 22-residue GS linker is only partially visible and is not involved in the PHTH–KD interaction. The total solvent-excluded surface area of the PH–KD interface is 753.8 Å$^2$. To further assess the importance of the unique PHTH/kinase interaction captured in this new structure we pursued mutational analysis combined with functional assays.

## Probing the functional role of PHTH/kinase interface

Using a coupled kinase assay, we tested the importance of R131, Y134, and R171 in regulating the activity of BTK. Since the R133/Y134 mutation was also the subject of the previous work examining the regulatory role of the PHTH domain (*Wang et al., 2015*), we examined single-point mutations (R133E, Y134E, and R171E), the double mutant studied previously (R133E/Y134E), and the triple mutation (R133E/Y134E/R171E) in an otherwise wild-type, active version of the PHTH–kinase construct ([1]PHTH[176]-G(GGSGG)$_4$G-[384]kinase[659]). BTK kinetics have been extensively characterized previously (*Dinh et al., 2007*), and our data show the same lag-phase and non-linear time course due to the activating effect of autophosphorylation on Y551 (*Figure 6a, d, g*). In an effort to approximate catalytic rates of the mutant and wild-type BTK proteins, we fit a line to the data within a short time window for each progress curve (*Figure 6b, e, h*) and compare time to a threshold value of Adenosine diphosphate produced in the assay (*Figure 6c, f, i*). The data for the PHTH–kinase construct show that neither the single mutations nor the double R133E/Y134E mutation alter the rate of the kinase reaction (*Figure 6b*) or the time to 3000 pmol ADP (*Figure 6c*). In contrast, mutation of all three PHTH domain side chains (R133E/Y134E/R171E) increases BTK activity (*Figure 6a, b*) and shortens the time to 3000 pmol ADP (*Figure 6c*). These data are consistent with the conclusion that the interface captured in the crystal structure serves an inhibitory role at least in the tethered PHTH–kinase complex.

We next tested the same PHTH mutations in the context of full-length BTK (*Figure 6d–f*). Comparing the activity of wild-type full-length BTK to the R133E/Y134E/R171E mutant, we find that disrupting the PHTH/kinase interface results in a slight increase in the activity of full-length BTK (*Figure 6d, e*). Since our new PHTH–kinase structures does not contain L390, and instead Y134 fulfills the hydrophobic stack (*Figure 5e*), we reasoned that in the full-length protein the PHTH domain may only exert its effect upon release of L390 from the N-lobe (likely concomitant with release of the SH3 domain). We therefore tested the effect of the PHTH mutations in the context of an additional mutation (W251K) that breaks the SH3 contacts with the SH2-linker (contains L390) and kinase domain N-lobe. Mutation of W251K in the SH3 domain of full-length BTK by itself leads to an increase in BTK activity similar to that of the R133E/Y134E/R171E mutation (*Figure 6d, e*). Mutation of W251 to lysine in combination with the R133E/Y134E/R171E mutation results in a more active kinase (*Figure 6d, e*) and shorter time to the threshold ADP level (*Figure 6f*) compared to constructs in which the autoinhibitory SH3 domain is intact. These data are consistent with the idea that the PHTH/kinase interaction and the SH3/SH2-linker/N-lobe contacts are mutually exclusive and that both interactions separately serve to allosterically inhibit the catalytic activity of the BTK kinase domain likely via stabilization of the hydrophobic stack.

## Effect of BTK PHTH mutations on activation by phospholipid

PH domains are well-characterized phospholipid-binding domains and the BTK PHTH domain has been studied extensively in this regard (*Chung et al., 2019*). PIP$_3$ binding promotes trans-autophosphorylation, presumably through formation of the PHTH Saraste dimer on the two-dimensional membrane surface. Given the involvement of the PHTH domain in membrane association and dimerization, we tested whether the activating mutation in the PHTH domain (R133E/Y134E/R171E/W251K) has any effect on PIP$_3$-mediated activation (*Figure 6g–i*). We find that wild-type BTK

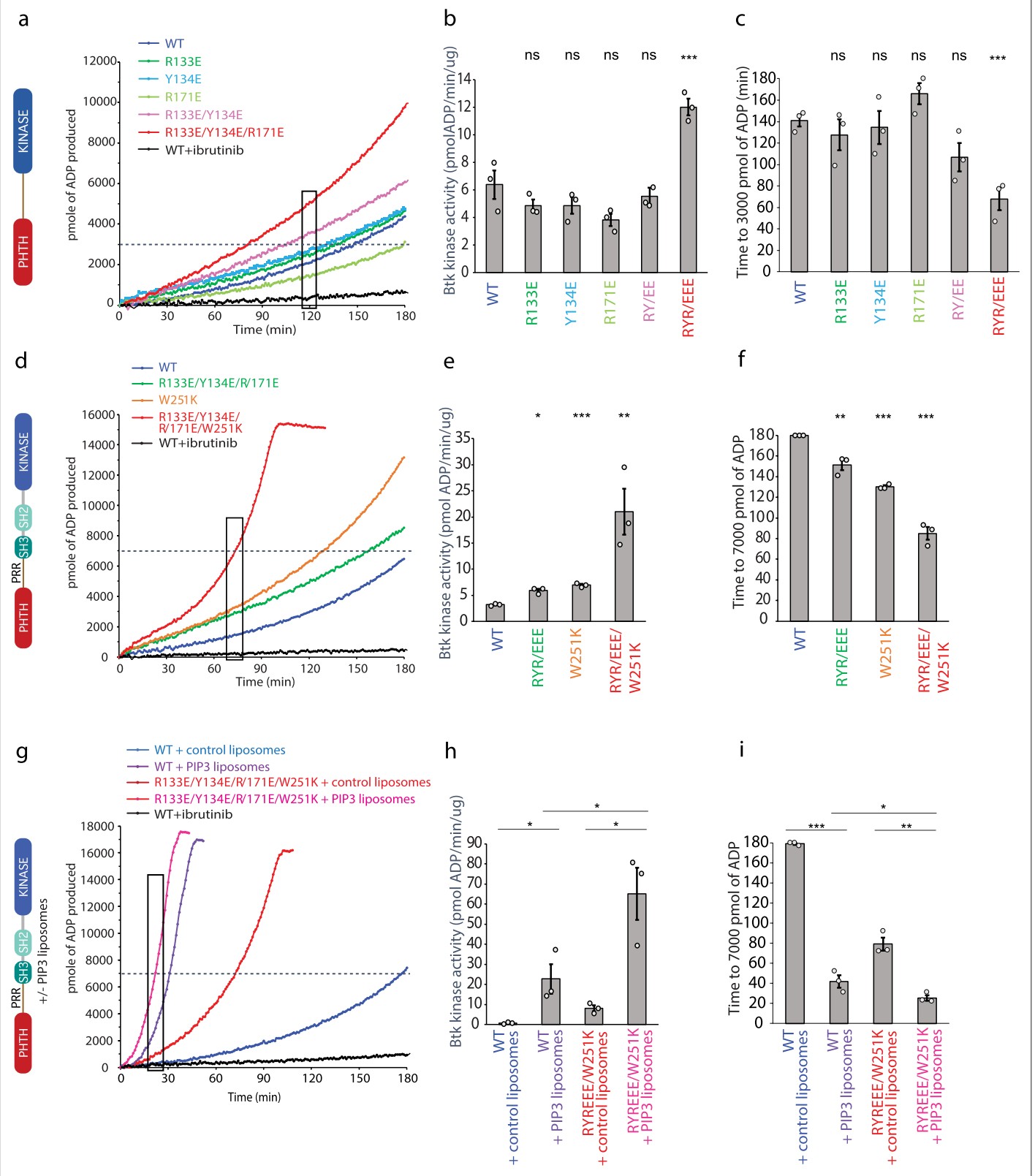

**Figure 6.** Bruton's tyrosine kinase (BTK) activity assays. (**a–c**) Representative progress curves, catalytic rate comparisons, and time to threshold ADP for the PH–KD construct. Wild-type BTK Pleckstrin homology/Tec homology (PHTH)–kinase protein is compared with single, double, and triple mutants to probe the PHTH/kinase interface. Ibrutinib inhibition leads to reduction of ADP production (black curve in all experiments). (**d–f**) Representative progress curves, catalytic rate comparisons, and time to threshold ADP for full-length BTK. Wild-type BTK is compared to the following full-length BTK

*Figure 6 continued on next page*

*Figure 6 continued*

mutants: R133E/Y134E/R171E, W251K, or W251K/R133E/Y134E/R171E. (**g–i**) Representative progress curves, catalytic rate comparisons, and time to threshold ADP for full-length WT BTK and R133E/Y134E/R171E mutant in the presence of either control or phosphatidylinositol (3,4,5)-trisphosphate (PIP$_3$) liposomes. (**a, d, g**) Representative progress curves of ADP production by BTK are from one of the three independent experiments, and each data point is the average of at least two replicates. Bar graphs (**b, e, h**) represent the average kinase activity rate ± standard error calculated from the boxed region of the corresponding progress curves. Bar graphs (**c, f, i**) represent the average time to a threshold value of ADP, indicated by dashed line on progress curves. Open circles on all bar graphs represent specific values in each independent experiment. For reactions for which the threshold ADP value is not reached (WT BTK in panels f and i) the values are reported as 180 min. The effect of mutations compared with the wild-type BTK was evaluated by Student's *t*-test (*p < 0.05; **p < 0.01; ***p < 0.001; ns, not significant).

and the active R133E/Y134E/R171E/W251K BTK mutant are both activated (*Figure 6g, h*) and reach threshold ADP levels more rapidly (*Figure 6i*) in the presence of PIP$_3$ liposomes compared to liposomes that do not contain PIP$_3$. Moreover, compared to wild-type BTK, the activating mutations in the PHTH domain (R133E/Y134E/R171E) result in increased activity even in the presence of PIP$_3$ (*Figure 6g, h*). Thus, the autoinhibitory effect of the PHTH domain is detected both in the context of non-membrane-associated BTK and when BTK is activated by PIP$_3$ binding to the PHTH domain.

## BTK kinase dimer structure

The focus of previous work examining BTK autophosphorylation at the PIP$_3$ containing membrane surface on has been on the PHTH/PIP$_3$ interaction and subsequent PHTH dimerization (*Chung et al., 2019*). PHTH-mediated dimerization at the membrane likely promotes a switch from the inactive to active kinase domain conformation followed by trans-autophosphorylation of the BTK kinase domain. Consistent with this activation model we have now crystallized the BTK kinase domain in a form that captures a dimer structure where activation loops are swapped to allow for trans-autophosphorylation.

The crystallization construct used to capture this new conformation of the BTK kinase domain included mutation of Y551E to mimic the phosphorylated activation loop, mutation of L390G to disfavor autoinhibitory contacts, and dasatinib bound to the active site (*Joseph et al., 2020*; *Marcotte et al., 2010*). The kinase domain forms a face-to-face dimer in the crystal (*Figure 7a*) with two intermolecular interfaces; the β3/C-helix loop in the N-lobe forms one interface (*Figure 7b*) and the domain swapped activation loop and G-helix mediate an extensive interface between the C-lobes (*Figure 7c*). In the N-lobe, the C-helix adopts an intermediate conformation between the previously solved active BTK kinase domain (PDB: 3K54) with the C-helix 'in' state and the inactive structure of the BTK kinase domain (PDB: 3GEN) with C-helix 'out'.

The activation loop of one monomer in the face-to-face BTK dimer structure extends into the active site of the other monomer but electron density is missing for residues 546–559, which includes the Y551 autophosphorylation site (*Figure 7d*). Nevertheless, superimposing two different substrate-bound kinase structures (tyrosine and serine phosphoacceptors) onto the BTK kinase domain (*Figure 7d*) shows that Y551 from one BTK kinase domain monomer would easily reach a viable phosphor-acceptor position in the active site of the opposing monomer. The swapped BTK dimer structure may therefore represent a conformation along the trans-autophosphorylation activation pathway (*Beenstock et al., 2016*).

The canonical Lys/Glu salt bridge with active kinase domains does not form in the BTK kinase domain dimer due to the unusual conformation of the phenylalanine side of the DFG motif (F540) (*Figure 7e*). The F540 side chain intercalates between the Lys/Glu side chains adopting what has previously been termed 'DFG-up' (*Dodson et al., 2010*; *Colombano et al., 2019*) or 'DFGinter' (*Modi and Dunbrack, 2019*). The C-helix is therefore held in an intermediate state between the C-helix in and C-helix out conformations associated with active and inactive kinases, respectively. In addition to the unusual position of F540, D539 of the DFG motif points inward, orientated approximately 180° opposite from that in the DFG-in conformation of the active kinase and E445 on the C-helix forms a hydrogen bond with the backbone amide of G541 in the DFG motif instead of forming a salt bridge with K430 (*Figure 7e*). It is also interesting to compare the regulatory spine (R-spine) for this BTK kinase dimer and the R-spine in previously solved active and inactive structures of BTK (*Figure 7f*). The active BTK structure shows close packing between the top of the R-spine and W395, which adopts the rotamer conformation required for active BTK (*Joseph et al., 2010*). Inactive BTK, in contrast, shows no contact between W395 and the R-spine (*Figure 7f*); interestingly, the DFG conformation is quite

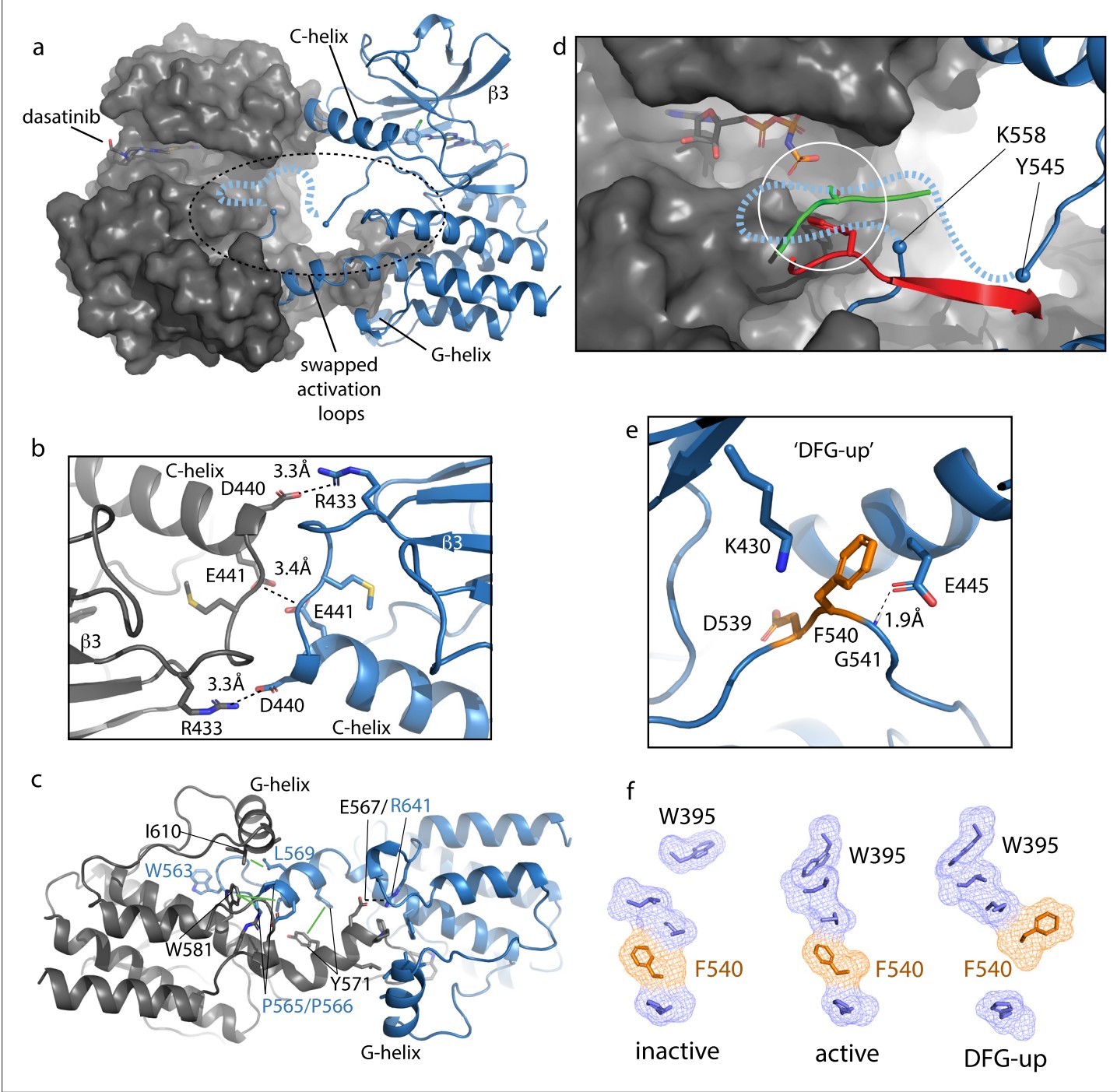

**Figure 7.** Bruton's tyrosine kinase (BTK) dimer. (**a**) Crystal structure of the BTK kinase domain dimer (PDB: 8S9F). Monomers are shown in gray and blue with one represented in cartoon and the other surface rendered. The C-helix, β3 strand, G-helix, and bound dasatinib are labeled. The region containing the swapped activation loops is indicated with a dashed circle. The portion of the activation loop for which electron density is missing is indicated with a dashed blue line. (**b, c**) Side chain interactions mediating the N- and C-lobe dimer interfaces, respectively. (**d**) Close-up view of activation loop of one monomer extending into the active site of the other monomer. Electron density is absent between Y545 and K558 (indicated with blue spheres). Dashed line indicates possible path for the 13 missing residues that contain Y551. Superimposed on the BTK dimer structure are the PKC kinase domain structure bound to substrate (PDB: 4DC2, green) and insulin-like growth factor 1 receptor kinase bound to substrate (PDB: 1K3A, red); kinase domains are excluded for clarity. The serine and tyrosine phosphoacceptors on these substrates are positioned close to the putative location of BTK Y551 (white circle). (**e**) Unusual 'DFG-up' conformation. In the BTK kinase dimer, F540 inserts between K430 and E445 preventing formation of the salt bridge associated with active kinases. (**f**) Comparison of regulatory spine structures for active BTK kinase domain (PDB: 3K54), inactive BTK (PDB: 3GEN), and

*Figure 7 continued on next page*

*Figure 7 continued*

the 'DFG-up' structure solved here (PDB: 8S9F). F540 is orange and other R-spine residues are in blue. W395 is at the top of the R-spine in BTK; the 'DFG-up' configuration stabilizes the active rotamer of W395 (*Joseph et al., 2010*; *Chopra et al., 2016*).

similar between active and inactive BTK. The 'DFG-up' configuration in the kinase domain dimer structure is accompanied by the W395 rotamer that is associated with active BTK (*Figure 7f*). That said, it is also possible that the DFG status is highly dependent on the nature of the bound drug. In fact, a previous structure of another TEC family kinase, BMX, reveals the same DFG-up state and the kinase domain in that case is also bound to dasatinib (*Muckelbauer et al., 2011*).

## Discussion

The findings reported here provide new insights into autoinhibition and allosteric control of the full-length, multidomain BTK protein. The data advance our understanding of BTK regulation as many previous studies did not interrogate the structure of the entire full-length protein. In pursuing full-length BTK, we find that missing electron density in crystals of full-length BTK, elongated SAXS envelopes, and visualization of the PHTH domain by cryoEM support a model where the N-terminal BTK PHTH domain populates a conformational ensemble surrounding the compact/autoinhibited SH3–SH2–kinase core (*Figure 8a*). In previous work probing allosteric regulation of full-length BTK (*Joseph et al., 2017*), we consistently find that perturbations to the conformational state of the SH3–SH2–kinase core have no effect on the hydrogen/deuterium exchange behavior of the PHTH domain consistent with the dynamically independent PHTH domain observed here. Contrary to models of autoinhibited multidomain proteins where each globular domain is sequestered into a compact core,

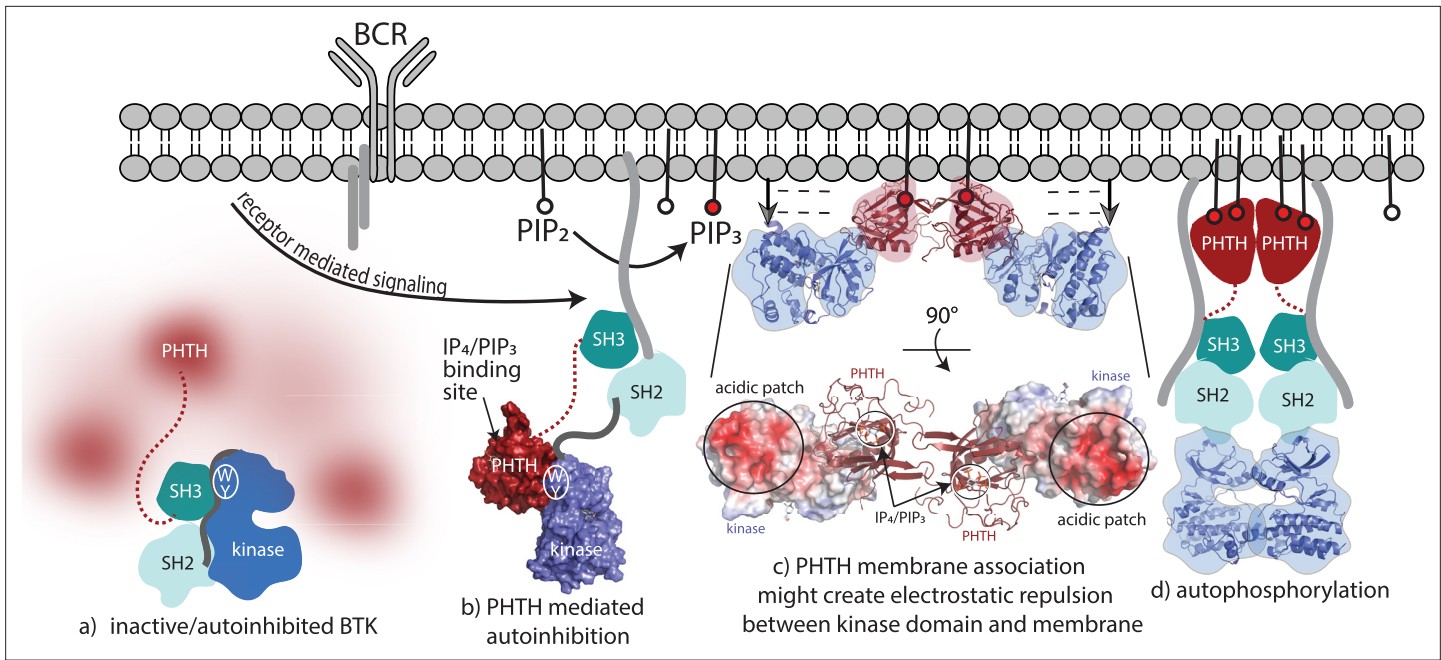

**Figure 8.** Bruton's tyrosine kinase (BTK) activation model. (**a**) Inactive, autoinhibited BTK, the conformational heterogeneity of the Pleckstrin homology/Tec homology (PHTH) domain is indicated in fuzzy red. The hydrophobic stack residues, W421 and Y461, are indicated on the kinase domain N-lobe as $W$ . (**b**) Engagement of the BTK SH3 and SH2 domains with exogenous ligands would allow for the PHTH domain to adopt its autoinhibitory pose. Surface rendering of structure solved here (PDB: 8S93) is included, accessibility of the inositol 1,3,4,5-tetrakisphosphate ($IP_4$)/phosphatidylinositol (3,4,5)-trisphosphate ($PIP_3$)-binding site is shown, and hydrophobic stack resides are indicated as in (**a**). (**c**) Two PHTH–kinase structures are superimposed on the Saraste PHTH dimer (PDB: 1B55). The top model indicates potential for electrostatic repulsion between membrane and BTK kinase domain in this arrangement (negative charges indicated by ----- and arrows suggest unfavorable interactions between negatively charged surfaces). A 90° rotation (bottom) shows the acidic patch on the kinase domain and the $PIP_3$-binding sites on the PHTH domain (circled) are on the same surface of the structure. (**d**) Release of all autoinhibitory contacts and dimerization of the BTK kinase domain (PDB: 8S9F) lead to autophosphorylation on Y551 in the activation loop of each kinase domain.

the autoinhibited state of full-length BTK might instead feature a conformationally heterogeneous PH domain that is connected to a compact core through a flexible linker and likely contacts the kinase domain only transiently. Similar conformational heterogeneity has been reported for the PH domains of phospholipases Cε and β (*Garland-Kuntz et al., 2018*).

Another PH domain containing kinase, AKT1, has been studied extensively (*Ashwell et al., 2012*; *Quambusch et al., 2019*; *Truebestein et al., 2021*; *Wu et al., 2010*; *Bae et al., 2022*; *Shaw et al., 2023*) and many of the structures that have been solved to elucidate autoregulation by the PH domain are seemingly biased by allosteric inhibitors. A recent crystal structure (*Truebestein et al., 2021*) provides a more unbiased view of the potential autoinhibitory interface between the PH and kinase domains of AKT; the PH domain binds to the activation loop face of the kinase domain in a manner that is quite similar to our earlier solution studies of both ITK and BTK (see *Figure 1e*; *Joseph et al., 2017*; *Devkota et al., 2017*; *Amatya et al., 2019*). Like AKT, the BTK PH/kinase interface detected in solution studies masks the PIP$_3$-binding site on the PH domain. However, consideration of all data to date suggests that regulatory interactions between the BTK PHTH and kinase domains are transient. Previously characterized contacts between the PHTH domain and the activation loop face (*Amatya et al., 2019*) or between the PHTH domain and the kinase domain N-lobe (*Wang et al., 2015*) may all exist within the conformational ensemble of a range PH/kinase domain interactions.

Non-PH domain containing kinases present similarly dynamic N-terminal regions that exert regulatory control over catalytic activity. The N-terminal region of the SRC kinases has been invoked in 'fuzzy' intramolecular regulatory interactions with the Src module (SH3–SH2–kinase) core (*Arbesú et al., 2017*). For SRC, it has been suggested that the intrinsically disordered N-terminal regions increase the flexibility of the SH3–SH2–kinase core to promote ligand-binding interactions (*Pérez et al., 2013*). The N-terminal 'cap' region of ABL is another example of a long, disordered N-terminal region; myristoylation results in a key autoinhibitory contact between the N-terminal myristate and the ABL kinase domain C-lobe (*Pluk et al., 2002*; *Nagar et al., 2003*; *Hantschel et al., 2003*; *Panjarian et al., 2013*). Within BTK, we have previously shown that the PRR in the linker between the PHTH and SH3 domains competes with the SH2–kinase linker for binding to the SH3 domain thereby shifting the conformational equilibrium of the autoinhibited core to a more open configuration (*Joseph et al., 2017*; *Laederach et al., 2002*). Indeed, BTK SH3 displacement from the autoinhibited core via interaction with the PRR may be a prerequisite for the PHTH domain to take up the autoinhibitory pose on the kinase domain N-lobe defined here (*Figure 8b*). The extent to which the intrinsically disordered PRR region of BTK participates in additional, perhaps fuzzy, interactions with the BTK SH3–SH2–kinase region remains to be investigated. As well, interactions of the N-terminal region of BTK with both positive and negative regulators (*Liu et al., 2001*; *Nocka et al., 2023*; *Tsukada et al., 1994*) may alter the dynamic characteristics observed here. It is also possible that the dynamic linker between the PHTH and SH3 domains of BTK may mediate phase separation (*Shelby et al., 2023*) and clustering into signaling competent condensates. While these questions need further study, it is notable that the SRC and TEC kinases seem to share dynamic regulatory features in their N-termini despite drastically different primary structures.

Transition from fully autoinhibited to active BTK following receptor activation can be considered in light of the structural insights now available (*Figure 8*). The BTK hydrophobic stack residues, W421 and Y461, are a binding site for both the L390 side chain on the SH2–kinase linker (*von Raußendorf et al., 2017*; *Figure 8a*) and Y134 on the BTK PHTH domain (*Figure 8b*). Both interactions allosterically inhibit BTK catalytic activity by stabilizing the inactive kinase domain conformation. The SH3/L390/hydrophobic stack interaction is present in the autoinhibited Src module and release of the regulatory SH3/SH2 domains opens the kinase N-lobe for secondary inhibitory interactions with the PHTH domain (*Figure 8b*). Once PIP$_3$ is made available by the action of PI3K, the PHTH binds to PIP$_3$ in the membrane and likely dimerizes through the Saraste dimer. To explore possible consequences of BTK dimerization at the membrane, we superimposed the PHTH–kinase structure onto the PHTH Saraste dimer (*Figure 8c*). Viewing the resulting dimer arrangement from the inner membrane surface shows a large acidic patch on the BTK kinase domains parallel to the bound IP$_4$ ligands. If the PHTH–kinase complex remains intact upon membrane association (*Figure 8c*), electrostatic repulsion between the negatively charged BTK kinase domain and the negatively charged membrane surface might result in dissociation of the inhibitory BTK PHTH/kinase interaction and subsequent dimerization-mediated autophosphorylation in the kinase domain (*Figure 8d*). Indeed, mutation of the PHTH domain

residues that contact the kinase domain led to more rapid PIP$_3$-mediated activation of full-length BTK (*Figure 6g–i*) suggesting the PHTH autoinhibitory structure interaction might remain intact as BTK associates with PIP$_3$. This is consistent with a recent report that Grb2 association with the PRR of BTK results in higher levels of BTK activity at the PIP$_3$ membrane possibly by destabilizing autoinhibited BTK (*Nocka et al., 2022*).

The autoinhibitory role of the PHTH domain in maintaining the BTK kinase domain in its inactive state until assembly at the membrane provides new insight into oncogenic forms of BTK. Specifically, the oncogenic p65BTK isoform lacks most of the PHTH domain and is expressed in colorectal carcinoma and some glioblastoma patients (*Grassilli et al., 2016*). As well, an extended transcript yields BTK-C, an isoform that contains additional residues at the N-terminus and is expressed in breast and prostate cancer cells (*Eifert et al., 2013*; *Wang et al., 2021*). It is not fully understood to what extent and through what mechanisms these deletions and extensions of the PHTH domain affect BTK regulation (*Grassilli et al., 2022*). In addition to effects on PIP$_3$ binding, and/or interactions with other regulatory proteins, it is also possible that the altered PHTH domain sequences may contribute to oncogenicity by disrupting the autoinhibitory function of the PHTH domain.

The TEC kinases provide a rich tapestry on which multidomain protein regulation and allostery can be studied. The observations here support a model where the BTK PHTH domain visits multiple conformational states and likely rapidly interconverts between states (both detached and associated with the SH3–SH2–kinase core). A model where the primary autoinhibitory contacts are housed within the SH3–SH2–kinase core is consistent with the fact that the SRC kinases lack the PH domain entirely and rely only on their SH3 and SH2 domains for autoinhibition (*Shah et al., 2018*). Despite the temptation to assign a stable autoinhibitory configuration to the BTK PHTH domain, the inherent conformational heterogeneity of the BTK PHTH domain instead suggests the possibility of multifaceted regulatory roles for this single domain. It will be interesting to compare the TEC kinases in this regard. The length of the PRR linker between PHTH and SH3 domain varies across the family, the loops that mediate formation of the BTK PHTH Saraste dimer are absent in the PHTH domains of other TEC kinases, and one TEC kinase, RLK/TXK, lacks the PHTH domain and instead contains an N-terminal region nearly identical in length to that of the SRC kinases. Subfamily-specific differences in structure, dynamics, autoregulation, activation, and allostery are likely important for control of distinct signaling pathways in distinct cell types.

## Materials and methods

**Key resources table**

| Reagent type (species) or resource | Designation | Source or reference | Identifiers | Additional information |
|---|---|---|---|---|
| Recombinant DNA reagent | FL BTK-C6H pET20 (plasmid) | DOI: 10.1016/j. str.2017.07.014 | | Residues 1–659 wlth Y617P mutation, UniProt: P35991 |
| Recombinant DNA reagent | FL BTK-C6H Q91A, I92A, I94A, and I95A pET20 (plasmid) | This paper | | Residues 1–659 wlth Y617P mutation |
| Recombinant DNA reagent | FL BTK-C6H A384P, S386P, T387P, and A388P pET20 (plasmid) | This paper | | Residues 1–659 wlth Y617P mutation |
| Recombinant DNA reagent | FL BTK-C6H A384P, S386P, T387P, A388P, and L390F pET20 (plasmid) | This paper | | Residues 1–659 wlth Y617P mutation |
| Recombinant DNA reagent | FL BTK-C6H L542M, S543T, V555T, R562K, S564A, and P565S pET20 (plasmid) | This paper | | Residues 1–659 wlth Y617P mutation |
| Recombinant DNA reagent | SH3-SH2-kinase BTK-C6H pET20 (plasmid) | This paper | | Residues 214–659 wlth Y617P mutation |
| Recombinant DNA reagent | SH3-SH2-kinase BTK-C6H A384P, S386P, T387P, A388P, and L390F pET20 (plasmid) | This paper | | Residues 214–659 wlth Y617P mutation |
| Recombinant DNA reagent | N6H-SUMO-FL BTK pET20 (plasmid) | This paper | | Residues 1–659 wlth Y617P mutation |

*Continued on next page*

*Continued*

| Reagent type (species) or resource | Designation | Source or reference | Identifiers | Additional information |
|---|---|---|---|---|
| Recombinant DNA reagent | N6H-SUMO-FL BTK W251K pET20 (plasmid) | This paper | | Residues 1–659 wIth Y617P mutation |
| Recombinant DNA reagent | N6H-SUMO-FL BTK R133E/Y134E/R171E pET20 (plasmid) | This paper | | Residues 1–659 wIth Y617P mutation |
| Recombinant DNA reagent | N6H-SUMO-FL BTK R133E/Y134E/R171E/W251K pET20 (plasmid) | This paper | | Residues 1–659 wIth Y617P mutation |
| Recombinant DNA reagent | N6H-SUMO-FL BTKsc pET20 (plasmid) | This paper | | Residues 1–659 wIth Y617P mutation |
| Recombinant DNA reagent | N6H-SUMO-FL BTKsc Δ185–194 pET20 (plasmid) | This paper | | Residues 1–184, 195–659 wIth Y617P mutation |
| Recombinant DNA reagent | N6H-SUMO-FL BTKsc Δ185–206 pET20 (plasmid) | This paper | | Residues 1–184, 207–659 wIth Y617P mutation |
| Recombinant DNA reagent | N6H-SUMO-FL BTKsc Δ181–206 pET20 (plasmid) | This paper | | Residues 1–180, 207–659 wIth Y617P mutation |
| Recombinant DNA reagent | N6H-SUMO-FL BTKsc Δ175–210 pET20 (plasmid) | This paper | | Residues 1–174, 211–659 wIth Y617P mutation |
| Recombinant DNA reagent | N6H-SUMO-FL BTKsc Δ173–215 pET20 (plasmid) | This paper | | Residues 1–172, 216–659 wIth Y617P mutation |
| Recombinant DNA reagent | N6H-SUMO-$^1$PHTH$^{171}$-G(GGSGG)$_4$G-$^{396}$kinase$^{659}$ Q91A, I92A, I94A, I95A, K430R, L542M, S543T, V555T, R562K, S564A, and P565S pET28 (plasmid) | This paper | | Residues 1–171, 396–659 wIth Y617P mutation |
| Recombinant DNA reagent | N6H-SUMO-$^1$PHTH$^{171}$-G(GGSGG)$_6$G-$^{396}$kinase$^{659}$ Q91A, I92A, I94A, I95A, K430R, L542M, S543T, V555T, R562K, S564A, and P565S pET28 (plasmid) | This paper | | Residues 1–171, 396–659 wIth Y617P mutation |
| Recombinant DNA reagent | N6H-SUMO-$^1$PHTH$^{176}$-G(GGSGG)$_4$G-$^{384}$kinase$^{659}$ pET28 (plasmid) | This paper | | Residues 1–176, 384–659 wIth Y617P mutation |
| Recombinant DNA reagent | N6H-SUMO-$^1$PHTH$^{176}$-G(GGSGG)$_4$G-$^{384}$kinase$^{659}$ R133E pET28 (plasmid) | This paper | | Residues 1–176, 384–659 wIth Y617P mutation |
| Recombinant DNA reagent | N6H-SUMO-$^1$PHTH$^{176}$-G(GGSGG)$_4$G-$^{384}$kinase$^{659}$ Y134E pET28 (plasmid) | This paper | | Residues 1–176, 384–659 wIth Y617P mutation |
| Recombinant DNA reagent | N6H-SUMO-$^1$PHTH$^{176}$-G(GGSGG)$_4$G-$^{384}$kinase$^{659}$ R133E/Y134E pET28 (plasmid) | This paper | | Residues 1–176, 384–659 wIth Y617P mutation |
| Recombinant DNA reagent | N6H-SUMO-$^1$PHTH$^{176}$-G(GGSGG)$_4$G-$^{384}$kinase$^{659}$ R133E/Y134E/171E pET28 (plasmid) | This paper | | Residues 1–176, 384–659 wIth Y617P mutation |
| Recombinant DNA reagent | N6H-SUMO KD Y551E/L390G pET20 (plasmid) | This paper | | Residues 382–659 wIth Y617P mutation |
| Recombinant DNA reagent | pCDFDuet YopH | Thomas E. Smithgall | | |
| Strain, strain background (*Escherichia coli*) | BL21 (DE3) | Thermo Fisher Scientific | Cat#C600003 | https://www.thermofisher.com/order/catalog/product/C600003 |
| Other | 1,2-Dioleoyl-*sn*-glycero-3-phosphocholine (DOPC) | Avanti Polar Lipids | Cat#850375P | https://avantilipids.com/product/850375 |
| Other | 1,2-Dioleoyl-*sn*-glycero-3-phospho-L-serine (DOPS) | Avanti Polar Lipids | Cat#840035P | https://avantilipids.com/product/840035 |

*Continued on next page*

*Continued*

| Reagent type (species) or resource | Designation | Source or reference | Identifiers | Additional information |
|---|---|---|---|---|
| Other | 1,2-Dioleoyl-*sn*-glycero-3-phosphoinositol-3,4,5-trisphosphate (tetra-ammonium salt) (PIP₃) | Avanti Polar Lipids | Cat#850156P | https://avantilipids.com/product/850156 |
| Other | Pyruvate Kinase/Lactic Dehydrogenase | MilliporeSigma | Cat#P0294 | https://www.sigmaaldrich.com/US/en/product/sigma/p0294?gclid=EAIaIQobChMIobDx8MqZgwMVyVdyCh1-egAIEAAYASAAEgLOzPD_BwE |
| Other | SYPRO Protein Gel Stains | Thermo Fisher Scientific | Cat#S6650 | https://www.thermofisher.com/order/catalog/product/S6650 |
| Other | Phusion Hot Start II DNA Polymerase | Thermo Fisher Scientific | Cat#F549L | https://www.thermofisher.com/order/catalog/product/F549L?SID=srch-srp-F549L |
| Antibody | BD Pharmingen Purified Mouse anti-Btk (pY551)/Itk (pY511) (mouse monoclonal) | BD Biosciences | Cat#558034 | WB (1:1000) |
| Software, algorithm | Coot | DOI: 10.1107/S0907444910007493 | v0.9.8.8 | https://www2.mrc-lmb.cam.ac.uk/personal/pemsley/coot/ |
| Software, algorithm | autoPROC | DOI: 10.1107/S0907444911007773 | v20211020 | https://www.globalphasing.com/autoproc/ |
| Software, algorithm | XDS | DOI: 10.1107/S0907444909047337 | v20220110 | https://xds.mr.mpg.de/ |
| Software, algorithm | STARANISO | DOI: 10.1107/S010876731809640X | v3.350 | https://staraniso.globalphasing.org/staraniso_about.html |
| Software, algorithm | CCP4i suite | DOI:10.1107/S2059798323003595 | v0.8 | https://www.ccp4.ac.uk/ |
| Software, algorithm | Phaser | DOI: 10.1107/S0021889807021206 | v2.8 | https://www.phaser.cimr.cam.ac.uk/index.php/Phaser_Crystallographic_Software |
| Software, algorithm | Phenix | DOI: 10.1107/S2059798319011471 | v1.20-4459 | https://phenix-online.org/ |
| Software, algorithm | CryoSparc | Structura Biotechnology Inc | v4.1.2 | https://cryosparc.com/ |
| Software, algorithm | PyMOL | Shrodinger LLC | v2.5.5 | https://www.pymol.org/ |
| Software, algorithm | DynamX | WATERS | v3.0 | https://www.waters.com/nextgen/us/en.html |
| Software, algorithm | PLGS | WATERS | v3.0 | https://www.waters.com/nextgen/us/en.html |
| Software, algorithm | FrameSlice | The SIBYLS Beamline | N/A | https://sibyls.als.lbl.gov/ran |
| Software, algorithm | BioXTAS RAW | DOI: 10.1107/S1600576717011438 | v2.1.4 | https://bioxtas-raw.readthedocs.io/en/latest/ |
| Software, algorithm | ATSAS: ALMERGE | EMBL Hamburg | v3.2.1 | https://www.embl-hamburg.de/biosaxs/manuals/almerge.html |
| Software, algorithm | Chimera | UCSF | v1.15 | https://www.cgl.ucsf.edu/chimera/ |

## Cloning and constructs

Cloning and mutagenesis in this work were carried by polymerase chain reaction (PCR) with Phusion Hot Start II DNA Polymerase (Thermo Fisher Scientific). For initial protein characterization, full-length (FL) BTK (residues 1–659) was cloned into pET20b vector with a C-terminal 6xHistidine tag (FL BTK-C6H) (*Joseph et al., 2017*). For crystallization and activity assays, a N-terminal 6× Histidine-SUMO-tag FL BTK (N6H-SUMO-FL BTK) construct was created by inserting a N-terminal 6× Histidine-SUMO-tag into pET20b FL BTK-C6H by Seamless Ligation Cloning Extract (SLiCE) cloning and removing the

C-terminal 6xHistidine by PCR (pET-His-SUMO provided by Eric Underbakke). The ${}^{1}$PHTH${}^{171}$-G(GGSGG)${}_{4/6}$G-${}^{396}$kinase${}^{659}$ constructs were cloned into pET28b vector with a N-terminal 6× Histidine-SUMO-tag. The BTK kinase domain (residues 382–659) construct (N6H-SUMO KD) was created by deleting residues 1–381 from the pET20b N6H-SUMO-FL BTK plasmid by PCR. All BTK constructs carry the solubilizing Y617P mutation (*Joseph et al., 2017*). All constructs created in this work were verified by sequencing at the Iowa State University DNA synthesis and sequencing facility.

## Protein expression and purification

The plasmids were transformed to BL21 (DE3) cells (MilliporeSigma). To express FL BTK-C6H, N6H-SUMO-FL BTK, and N6H-SUMO KD, cells were grown in LB broth (Fisher Scientific) supplemented with 100 µg/ml ampicillin (Fisher Scientific). For the expression of ${}^{1}$PHTH${}^{171}$-G(GGSGG)${}_{4}$G-${}^{396}$kinase${}^{659}$ and ${}^{1}$PHTH${}^{176}$-G(GGSGG)${}_{4}$G-${}^{384}$kinase${}^{659}$ fusion proteins (PHTH–KD), cells were grown in LB broth supplemented with 50 µg/ml kanamycin (MilliporeSigma). For expression of any kinase active BTK, pCDFDuet YopH was co-transformed with the BTK plasmids additionally supplemented with 50 µg/ml streptomycin. All BTK constructs were expressed similarly: cells were grown at 37°C until the optical density at 600 nm reached ~0.6. Protein expression was induced with 1 mM isopropyl β-D-1-thiogalactopyranoside for 24 hr at 18°C. Cells were harvested by centrifugation at 3000 × *g* for 20 min, resuspended in lysis buffer (50 mM Tris pH 8.0, 500 mM NaCl and 10% glycerol), and stored at −80°C until use. Cells were thawed with the addition of 0.5 mg/ml lysozyme and 1500 U of DNAse I (MilliporeSigma) and lysed by sonication on ice at 50% amplitude for 2 min with 0.5 s on and 1.5 s off intervals. The lysate was clarified by centrifugation at 20,000 × *g* for 1 hr at 4°C, and the supernatant was applied to a Ni-NTA column (QIAGEN) equilibrated with lysis buffer. The column was washed with 20 column volume of lysis buffer with 10 mM imidazole. For FL BTK-C6H, the proteins eluted with lysis buffer with 300 mM imidazole. For N6H-SUMO-tagged BTK constructs, the BTK-bound Ni-NTA resins in disposable columns were capped, resuspended with the lysis buffer with the addition of 5 mM Tris(2-carboxyethyl)phosphine and 1 mg of Ulp1 (provided by Eric Underbakke), and incubated O/N. The tagless BTK proteins were obtained by collecting the flow-through from the column and eluting with additional 10 ml of lysis buffer. The BTK sample was concentrated, and further purified with HiLoad 26/600 Superdex 200 pg (Cytiva) with Size exclusion chromatography buffer (20 mM Tris pH 8.0, 150 mM NaCl, 10% glycerol, and 1 mM Dithiothreitol). The purity of the samples was verified by SDS–PAGE. If YopH is still present in the BTK fractions, proteins were further purified with a Source 15Q column using ion-change buffers (Buffer A: 20 mM Tris pH 8.0, 1 DTT, and 50 mM; Buffer B: 20 mM Tris pH 8.0, 1 DTT, and 1 M NaCl). Purified proteins were concentrated to 5–20 mg/ml, flash-frozen in 100 µl aliquots with liquid nitrogen, and stored at −80°C.

## Crystallization, structure determination, and refinement

### Crystal structure of FL BTK

The FL BTK construct (residues 1–659) used for initial crystallization trials contains multiple mutations in different regions of BTK: (1) PHTH domain β5–6 loop mutations: Q91A, I92A, I94A, and I95A; (2) PPR region mutations: P189A, P192A, P203A, and P206A; (3) SH2–kinase linker mutations: A384P, S386P, T387P, A388P, and L390F; (4) activation loop mutations: L542M S543T, V555T, R562K, S564A, and P565S; (5) kinase inactive mutation: K430R. Tiny needle-shaped crystals of FL BTK were initially obtained from sitting-drop vapor diffusion method by mixing purified FL BTK proteins at 15–20 mg/ml with equal amount of the precipitant solution 15–20% polyethylene glycol (PEG) 3350, 0.1 M Bis-Tris Propane, pH7.5, and 0.2 M potassium thiocyanate (KSCN) at 4°C. Larger FL BTK crystals (0.2–0.4 mm) were obtained after E298A, K300A, and E301A mutations were introduced to the construct based on results from Surface Entropy Reduction prediction (SERp) server (*Goldschmidt et al., 2007*). Final FL BTK construct for crystallization (FL BTKsc) contains only the SH2–KD linker mutations (A384P, S386P, T387P, A388P, and L390F), activation loop mutations (L542M S543T, V555T, R562K, S564A, and P565S; also included in previous solution studies mapping the PHTH/kinase interaction *Amatya et al., 2019*); active kinase salt bridge mutation (K430R) and the SERp mutations (E298A, K300A, and E301A). In an effort to reduce the flexibility of the PRR containing linker, five additional full-length BTK constructs with the following deletions: Δ185–194, Δ185–206, Δ181–206, Δ175–210, and Δ 173–215 were also subject to crystallization and structural analysis, but no electron density was observed for the PHTH domain from any of these crystals.

For X-ray diffraction data collection, crystals were transferred, with six sequential steps with increasing amount of glycerol, to a solution containing 17% PEG 3350, 0.1 M Bis-Tris Propane, pH7.5, 0.2 M KSCN, and 20% glycerol, and then flash-frozen with liquid nitrogen. X-ray diffraction data were collected at the Advanced Photon Source (APS) beamline 23-ID-B GM/CA and 24-ID-E NE-CAT. Datasets were indexed, merged, and scaled using autoPROC (Global Phasing) (*Vonrhein et al., 2011*; *Tickle et al., 2018*). The structure was solved by molecular replacement (MR) using Phaser in the CCP4i suite (*Winn et al., 2011*; *McCoy et al., 2007*) using the previous SH3–SH2–KD structure (PDB code 4XI2) as the search model. The structures were refined using Phenix (*Liebschner et al., 2019*) and built in Coot (*Emsley and Cowtan, 2004*). Complete X-ray collection and refinement statistics are provided in *Table 1*. The figures are prepared with Pymol (*DeLano, 2002*).

## Crystal structure of the PHTH–KD complex

PHTH–KD proteins at 10 mg/ml were crystalized with hanging-drop method by mixing at 1:1 ratio with precipitant solution, 20% PEG 3350, 0.1 M Bis-Tris, pH 5.6, 0.2 M MgCl$_2$. Thin plate-like crystal clusters were harvested and flash-frozen in the cryo-protectant solution (precipitant solution + 20% glycerol). X-ray diffraction data were collected at the APS beamline 23-ID-B GM/CA. PH–KD structure was solved by MR with the isolated structures of the PHTH domain (PDB code 1BTK) and the kinase domain bound to GDC-0853 (PDB code 5vfi) (*Hyvönen and Saraste, 1997*; *Crawford et al., 2018*). The structures were refined using Phenix (*Liebschner et al., 2019*) and built in Coot (*Emsley and Cowtan, 2004*).

## Crystal structure of the kinase domain bound to dasatinib

BTK KD Y551E/L390G (residue 382–659) at 20 mg/ml was crystalized with the sitting-drop method by mixing at 1:1 ratio with precipitant solution, 42% PEG 200, 0.1 M 4-(2-hydroxyethyl)piperazine-1-ethane-sulfonic acid, pH 7.5. Crystals were harvested and flash-frozen in the cryo-protectant solution (precipitant solution + 20% glycerol). X-ray diffraction data were collected at the APS beamline 23-ID-B GM/CA and 24-ID-E NE-CAT. The structure of BTK KD bound to dasatinib was solved by MR with the previous dasatinib-bound KD structure (PDB code 3K54) (*Marcotte et al., 2010*). The structures were refined using Phenix (*Liebschner et al., 2019*) and built in Coot (*Emsley and Cowtan, 2004*).

All structures were deposited to the Protein Data Bank with accession codes as follows: FL BTK: 8GMB; PHTH-KD: 8S93; BTK KD/dasatinib: 8S9F.

## CryoEM sample preparation, data collection, and processing

An aliquot of purified FL BTK (18 mg/ml) was thawed and re-purified with a Superdex 200 10/300 GL column. The peak fractions were either used directly for cryoEM grid preparation or flash-frozen with liquid nitrogen. 3 µl of FL BTK at 0.4–0.6 mg/ml were applied to freshly glow-discharged 300-mesh Quantifoil R1.2/1.3 copper grids (EMS), blotted for 3 s with the blot force of 3, and plunged into liquid ethane using FEI Mark IV Vitrobot operated at 4°C and 100% humidity. Dataset was collected on a 200 kV Glacios Cryo-TEM electron microscope (Thermo Fisher Scientific) equipped with a Gatan K3 direct electron detector in super resolution mode using Serial EM at a pixel size of 0.45315 Å with total dose of 59 electrons Å$^{-2}$ distributed over 50 frames. 755 movies were selected for data processing. Movies were binned 2× to 0.9063 Å during alignment using Patch Motion Correction. CTF was estimated using Patch CTF estimation and particles were picked by blob picker in Cryosparc-Live. All further steps were done in Cryosparc v4.1.2 (*Punjani et al., 2017*). Picked particles were extracted with a box size of 386 pix and fourier cropped to a box size of 184 pix. Extracted particles were subjected to multiple rounds of 2D classification resulting in 68,788 particles which were classified into four ab initio reconstructions and further refined using non-uniform refinement (*Punjani et al., 2020*). All maps were visualized and analyzed using UCSF Chimera (*Pettersen et al., 2004*). Electron Microscopy Data Bank (EMDB) accession codes: EMD-40585, EMD-40586, and EMD-40587.

## Hydrogen deuterium exchange mass spectrometry

Prior to deuterium labeling, wild-type, full-length BTK (20 µM) was thawed on ice and then incubated at 23°C (room temperature) for 1 hr, diluted to 8 µM with buffer (20 mM Tris pH 8.0, 150 mM NaCl, 10% glycerol) and then returned to ice in preparation for deuterium labeling. Deuterium labeling was initiated with an 18-fold dilution (36 µl) into D$_2$O buffer (20 mM Tris pD 8.0, 150 mM NaCl, 10%

**Table 1.** Data collection and refinement statistics.

| | FL Btk[†,‡] | PHTH/KD complex[*,‡] | KD with dasatinib[†,‡] |
|---|---|---|---|
| PDB codes | 8GMB | 8S93 | 8S9F |
| **Data collection** | | | |
| Space group | $P\,3_1\,2\,1$ | $P\,1\,2_1\,1$ | $P\,1\,2_1\,1$ |
| Cell dimensions | | | |
| $a, b, c$ (Å) | 125.72, 125.72, 110.09 | 38.38, 77.38, 82.32 | 55.24, 110.04, 61.14 |
| $\alpha, \beta, \gamma$ (°) | 90, 90, 120 | 90, 97.98, 90 | 90, 99.51, 90 |
| Resolution (Å)[§] | 108.88–3.19 (3.53–3.19) | 81.53–2.00 (2.20–2.00) | 55.02–2.50 (2.79–2.50) |
| Spherical data completeness (%)[§] | 67.7 (13.2) | 54.5 (13.6) | 69.7 (12.5) |
| Ellipsoidal data completeness (%)[§] | 94.2 (70.4) | 84.1 (78.5) | 91.8 (52.0) |
| $R_{merge}$[§] | 0.27 (>1) | 0.13 (0.80) | 0.26 (>1) |
| $R_{meas}$[§] | 0.27 (>1) | 0.16 (0.95) | 0.396 (>1) |
| $CC_{1/2}$[§] | 0.998 (0.74) | 0.99 (0.54) | 0.99 (0.56) |
| Mean $I/\sigma$[§] | 17.8 (2.0) | 7.1 (1.6) | 11.2 (1.4) |
| Multiplicity[§] | 88.5 (82.2) | 3.7 (3.3) | 14.8 (14.9) |
| **Refinement** | | | |
| Resolution (Å)[§] | 19.58–3.40 (3.52–3.40) | 19.71–2.10 (2.18–2.10) | 19.96–2.60 (2.69–2.60) |
| Number of reflections[§] | 11,205 (295) | 21,392 (572) | 17,286 (300) |
| $R_{work}/R_{free}$ (%) | 27.63/28.93 | 19.95/24.86 | 24.34/28.95 |
| Number of non-hydrogen atoms | 3168 | 3759 | 4167 |
| Protein | 3119 | 3525 | 4034 |
| Ligand/ion | 93 | 108 | 118 |
| Water | 0 | 178 | 67 |
| Average $B$-factor | 157.69 | 39.89 | 78.30 |
| Protein | 158.15 | 40.15 | 78.73 |
| Ligand/ion | 128.60 | 36.73 | 74.73 |
| Water | | 35.74 | 56.37 |
| Ramachandran statistics | | | |
| Favored (%) | 93.72 | 97.62 | 96.59 |
| Allowed (%) | 6.05 | 2.38 | 3.41 |
| Outliers (%) | 0.23 | 0.00 | 0.00 |
| Clashscore | 6.56 | 1.69 | 3.27 |
| R.m.s. deviations | | | |
| Bond lengths (Å) | 0.002 | 0.002 | 0.002 |
| Bond angles (°) | 0.43 | 0.49 | 0.43 |

[*]X-ray data from a single crystal.

[†]X-ray data from multiple crystals.

[‡]X-ray data anisotropically corrected with the STARANISO webserver (Global Phasing).

[§]Statistics for the highest resolution shell are shown in parentheses.

glycerol, 99.9% $D_2O$). After each labeling time (10 s, 1 min, 10 min, 1 hr, 4 hr) at room temperature, the labeling reaction was quenched with the addition of an equal volume (38 µl) of ice-cold quenching buffer (150 mM potassium phosphate, pH 2.4, $H_2O$) and analyzed immediately.

Deuterated and control samples were digested online at 15°C using an Enzymate pepsin column. The cooling chamber of the HDX system, which housed all the chromatographic elements, was held at 0.0 ± 0.1°C for the entire time of the measurements. Peptides were trapped and desalted on a VanGuard Pre-Column trap (2.1 mm × 5 mm, ACQUITY UPLC BEH C18, 1.7 µm) for 3 min at 100 µl/ min. Peptides were eluted from the trap using a 5–35% gradient of acetonitrile with 0.1% formic acid over 6 min at a flow rate of 100 µl/min, and separated using an ACQUITY UPLC HSS T3, 1.8 µm, 1.0 mm × 50 mm column. Mass spectra were acquired using a Waters Synapt G2-Si HDMS[E] mass spectrometer in ion mobility mode. The error of determining the average deuterium incorporation for each peptide was at or below ±0.25 Da, based on deuterated peptide standards.

Peptides were identified from replicate HDMS[E] analyses (as detailed in the Supplemental Data file) of undeuterated control samples using PLGS 3.0.3 (Waters Corporation). The peptides identified in PLGS were then filtered in DynamX 3.0 (Waters Corporation) implementing a minimum products per amino acid cut-off of 0.25, at least two consecutive product ions (see Supplemental Data file). Those peptides meeting the filtering criteria to this point were further processed by DynamX 3.0 (Waters Corporation). The relative amount of deuterium in each peptide was determined by the software by subtracting the centroid mass of the undeuterated form of each peptide from the deuterated form, at each time point. Deuterium levels were not corrected for back exchange and thus reported as relative (**Wales and Engen, 2006**).

All deuterium uptake values are provided in the Supplemental Data file along with an HDX data summary and comprehensive list of experimental parameters in the recommended (**Masson et al., 2019**) tabular format. All HDX MS data have been deposited to the ProteomeXchange Consortium via the PRIDE partner repository (**Perez-Riverol et al., 2022**) with the dataset identifier PXD041657.

## Thermal shift assay

Purified proteins at 0.5 mg/ml were mixed with SYPRO Protein Gel Stains (Thermo Fisher Scientific) at 1:500 dilution. 20 µl of the protein/SYPRO mixture were aliquoted to MicroAmp Fast Optical 96-Well Reaction Plate (Thermo Fisher Scientific). The experiments with temperature ramp from 25 to 95°C and 0.5°C/step were measured with The Applied Biosystems StepOnePlus 96-well qPCR system (Thermo Fisher Scientific). The melt curve data were analyzed with the StepOne Software (Thermo Fisher Scientific) and the melting temperatures of each construct were determined by taking the lowest point of the negative derivative of normalized fluorescence. Three measurements from each mutant were normalized, averaged, and plotted with Excel.

## Preparation of liposomes

1,2-Dioleoyl-*sn*-glycero-3-phosphocholine (DOPC), 1,2-dioleoyl-*sn*-glycero-3-phospho-L-serine (DOPS), and 1,2-dioleoyl-*sn*-glycero-3-[phosphoinositol-3,4,5-trisphosphate] (tetra-ammonium salt) ($PIP_3$) (Avanti Polar Lipids) were dissolved in chloroform at 10 mg/ml and stored at −20°C. To prepare control and $PIP_3$ liposomes, DOPC, DOPS, and $PIP_3$ were mixed in glass test tubes at the molar ratio of 80:20:0 and 75:20:5 (DOPC:DOPS:$PIP_3$), respectively. Chloroform was removed by blowing a gentle stream of nitrogen gas until no solvent is visible. The lipids were transferred to a vacuum desiccator and dried overnight at room temperature. Dry lipid films were hydrated in a buffer containing 50 mM HEPES pH 7.4, 150 mM NaCl, and 5% glycerol to a final concentration of 12.5 mM. The hydrated liposomes were subjected to three freeze–thaw cycles using liquid nitrogen, followed by 10 passes through 100 nm filters (MilliporeSigma).

## NADH-coupled kinase assay

The kinase activity of BTK was determined using pyruvate kinase (PK)/lactate dehydrogenase (LDH) coupled kinase assay (**Barker et al., 1995**). 5 µl of BTK at 20 µM or 5 µl of SEC buffer (for blanks) was added to 20 µl of the reaction buffer containing 50 mM HEPES, pH 7.4, 150 mM NaCl, 5% glycerol, 1 mM DTT, and 1.6 µl of PK/LDH enzymes from rabbit muscle (MilliporeSigma). For the liposome experiments, 200 µM liposomes were added to the both control and kinase reactions. Experiments were performed in triplicates in black 96-well plates and were measured alongside three control reactions

for nonenzymatic production of ADP (no kinase added). The reactions were initiated by adding 25 µl of 2× ATP/NADH buffer, containing 1 mM NADH, 2 mM PEP, 2 mM ATP and 20 mM MgCl$_2$ in reaction buffer and measured using the Synergy HT Microplate Reader (BioTek) by monitoring the fluorescence of NADH at 460 nm at 22°C with 1 min intervals for 3 hr. The absolute concentration of NADH stock at ~40 mM was determined by measuring its absorbance at 340 nm ($\varepsilon$340 = 6220 M$^{-1}$ cm$^{-1}$) based on Beer–Lambert law. NADH concentration versus fluorescence units (FUs) standard curve ranging from 50 to 600 µM was measured at beginning of each measurement to determine the FUs to NADH concentration conversion factor. Raw fluorescence data from control reactions and kinase reactions for each condition were averaged and converted to picomole ADP produced using the NADH concentration conversion factor calculated assuming the amount of NDAH consumed is equivalent to the amount of ADP produced by ATP hydrolysis. Amount of ADP produced corrected for nonenzymatic ADP production (from control reactions) were plotted against time in Excel. The reaction rate of each experiment was calculated from the slope of the curve in the boxed regions.

## Kinase assay and western blotting

Kinase assays were performed by incubating 1 µM full-length BTK in a buffer containing 50 mM HEPES pH 7.4, 150 mM NaCl, 5% glycerol, 1 mM DTT, 10 mM MgCl$_2$, and 1 mM ATP. Activities of wild-type and mutant BTK proteins were carried out at room temperature. Samples from the reactions were collected and quenched at different time points by mixing with 4× SDS sample buffer for western blotting with the anti-pY551 antibody (BD Biosciences). The samples from 0 min were analyzed by SDS–PAGE as loading controls. Samples from 5, 10, and 20 min were analyzed by western blotting. The blots were quantified using the ChemiDoc detection system (Bio-Rad).

## SAXS

Purified full-length BTK protein and the BTK SH3–SH2–kinase fragment (both with C-terminal hexa-His tags) at 10 mg/ml were dialyzed against a buffer containing 20 mM Tris, 150 mM NaCl, 1 mM DTT, and 2% glycerol. Four different concentrations of samples (1, 2, 5, and 10 mg/ml) were shipped to the MailinSAXS offered by the SIBYLS Beamline at the Advanced Light Source in Berkeley, California. The data were collected with X-ray wavelength at 1.27 Å and the sample-to-detector distance at 2 m. Each sample was exposed to the beam for 10 s with data collected as 0.2 s slices, resulting total 50 frames of diffraction data per sample. Collected data were processed using the SIBYLS SAXS FrameSlice server (https://sibyls.als.lbl.gov/ran). ALMERGE was used to combine and extrapolate SAXS datasets collected at different concentrations to infinite dilution (*Franke et al., 2012*). Merged data were analyzed with Raw (*Hopkins et al., 2017*). Ab initio reconstruction were performed with DAMMIF/N through Raw (*Franke and Svergun, 2009*). Reconstructed bead models were visualized with Chimera. Small Angle Scattering Biological Data Bank (SASBDB) accession codes: SASDRB9, SASDRC9, SASDRD9, and SASDRE9.

## Acknowledgements

This work is supported by a grant from the National Institutes of Health (National Institute of Allergy and Infectious Diseases, AI43957) to AHA We also thank the Roy J Carver Charitable Trust, Muscatine, Iowa for ongoing research support and investment in the Iowa State University (ISU) Cryo-EM facility; data were collected on the Glacios 200 kV TEM and Gatan K3 detector in this facility. Charles Stewart and the ISU Macromolecular X-ray Crystallography Facility arranged for shipping crystals and data collection. Greg Hura and the staff of the SIBYLS beamline at the Advanced Light Source provided training in SAXS data processing, interpretation, and data presentation for publication. ALS is a national user facility operated by Lawrence Berkeley National Laboratory on behalf of the Department of Energy, Office of Basic Energy Sciences, through the Integrated Diffraction Analysis Technologies (IDAT) program, supported by DOE Office of Biological and Environmental Research. Additional support comes from the National Institutes of Health project ALS-ENABLE (P30 GM124169) and a High-End Instrumentation Grant S10OD018483. This research also used resources of GM/CA and NE-CAT beamlines at the Advanced Photon Source. GM/CA has been funded by the National Cancer Institute (ACB-12002), the National Institute of General Medical Sciences (AGM-12006, P30GM138396), and the NIH shared instrumentation grant (S10 OD012289). NECAT has been funded by the National Institutes of Health (P30 GM124165) and a NIH-ORIP HEI grant (S10OD021527).

## Additional information

### Competing interests

Amy H Andreotti: Senior editor, eLife. The other authors declare that no competing interests exist.

### Funding

| Funder | Grant reference number | Author |
|---|---|---|
| National Institutes of Health | R01AI043957 | Amy H Andreotti |

The funders had no role in study design, data collection, and interpretation, or the decision to submit the work for publication.

### Author contributions

David Yin-wei Lin, Conceptualization, Investigation, Writing - original draft; Lauren E Kueffer, Thomas E Wales, Investigation; Puneet Juneja, Formal analysis; John R Engen, Supervision; Amy H Andreotti, Conceptualization, Supervision, Funding acquisition, Project administration, Writing – review and editing

### Author ORCIDs

David Yin-wei Lin ⓘ http://orcid.org/0000-0001-6149-5236
Thomas E Wales ⓘ http://orcid.org/0000-0001-6133-5689
John R Engen ⓘ http://orcid.org/0000-0002-6918-9476
Amy H Andreotti ⓘ https://orcid.org/0000-0002-6952-7244

Reviewer #1 (Public Review): https://doi.org/10.7554/eLife.89489.3.sa1
Reviewer #2 (Public Review): https://doi.org/10.7554/eLife.89489.3.sa2
Reviewer #3 (Public Review): https://doi.org/10.7554/eLife.89489.3.sa3
Author Response https://doi.org/10.7554/eLife.89489.3.sa4

## Additional files

### Supplementary files

• Supplementary file 1. Summary and list of experimental parameters for hydrogen deuterium exchange data acquisition.
• MDAR checklist

### Data availability

Diffraction data have been deposited in PDB under the accession codes 8S93, 8GMB, 8S9F.

The following datasets were generated:

| Author(s) | Year | Dataset title | Dataset URL | Database and Identifier |
|---|---|---|---|---|
| Lin DY, Andreotti AH | 2023 | Crystal structure of the PH-TH/kinase complex of Bruton's tyrosine kinase | https://www.rcsb.org/structure/8S93 | RCSB Protein Data Bank, 8S93 |
| Lin DY, Andreotti AH | 2023 | Crystal structure of the full-length Bruton's tyrosine kinase (PH-TH domain not visible) | https://www.rcsb.org/structure/8GMB | RCSB Protein Data Bank, 8GMB |
| Lin DY, Andreotti AH | 2023 | Crystal structure of the kinase domain of Bruton's Tyrosine Kinase bound to dasatinib | https://www.rcsb.org/structure/8S9F | RCSB Protein Data Bank, 8S9F |

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
