## [Editor Report · eLife assessment]

BTK, a TEC-family tyrosine kinase activated by the B-cell antigen receptor, contains a variety of regulatory domains and it is subject to complex regulation by membrane phospholipids, protein ligands, phosphorylation, and dimerization. This study presents **convincing** evidence, utilizing various biophysical techniques, to support a model for BTK activation that will be **valuable** for the field. Overall, the study enhances the understanding of BTK's activation mechanism, autoinhibition, and allosteric control, challenging previous assumptions about BTK.

---

## [Referee Report · Reviewer #1 (Public Review)]

The manuscript by Lin et al describes a wide biophysical survey of the molecular mechanisms underlying full length BTK regulation. This is a continuation of this lab's excellent work on deciphering the myriad levels of regulation of BTKs downstream of their activation by plasma membrane localised receptors.

The manuscript uses a synergy of cryo EM, HDX-MS and mutational analysis to delve into the role of the how the accessory domains modify the activity of the kinase domain. The manuscript essentially has three main novel insights into BTK regulation.

1. Cryo EM and SAXS shows that the PHTH region is dynamic compared to the conserved Src module.

2. A 2nd generation tethered PH-kinase construct crystal of BTK reveals a unique orientation of the PH domain relative to the kinase domain, that is different from previous structures.

3. A new structure of the kinase domain dimer shows how trans-phosphorylation can be achieved.

Excitingly these structural work allow for the generation of a model of how BTK can act as a strict coincidence sensor for both activated BCR complex as well as PIP3 before it obtains full activity. To my eye the most exciting result of this work is describing how the PH domain can inhibit activity once the SH3/SH2 domain is disengaged, allowing for an additional level of regulatory control.

I have very few experimental concerns as the methods and figures are well described and clear. As the authors are potentially saying that the previously solved PH domain-kinase interface is but one of many possible inhibitory conformations that can be adapted.

---

## [Referee Report · Reviewer #2 (Public Review)]

In this study, multiple biophysical techniques were employed to investigate the activation mechanism of BTK, a multi-domain non-receptor protein kinase. Previous studies have elucidated the inhibitory effects of the SH3 and SH2 domains on the kinase and the potential activation mechanism involving the membrane-bound PIP3 inducing transient dimerization of the PH-TH domain, which binds to lipids.

The primary focus of the present study was on three new constructs: a full-length BTK construct, a construct where the PH-TH domain is connected to the kinase domain, and a construct featuring a kinase domain with a phosphomimetic at the autophosphorylation site Y551. The authors aimed to provide new insights into the autoinhibition and allosteric control of BTK.

The study reports that SAXS analysis of the full-length BTK protein construct, along with cryoEM visualization of the PH-TH domain, supports a model in which the N-terminal PH-TH domain exists in a conformational ensemble surrounding a compact/autoinhibited SH3-SH2-kinase core. This finding is interesting because it contradicts previous models proposing that each globular domain is tightly packed within the core.

Furthermore, the authors present a model for an inhibitory interaction between the N-lobe of the kinase and the PH-TH domain. This model is based on a study using a tethered complex with a longer tether than a previously reported construct where the PH-TH domain was tightly attached to the kinase domain (ref 5). The authors argue that the new structure is relevant. However, this assertion requires further explanation and discussion, particularly considering that the functional assays used to assess the impact of mutating residues within the PH-TH/kinase domain contradict the results of the previous study (ref 5).

Additionally, the study presents the structure of the kinase domain with swapped activation loops in a dimeric form, representing a previously unseen structure along the trans-phosphorylation pathway. This structure holds potential relevance. To better understand its significance, employing a structure/function approach like the one described for the PH-TH/kinase domain interface would be beneficial.

Overall, this study contributes to our understanding of the activation mechanism of BTK and sheds light on the autoinhibition and allosteric control of this protein kinase. It presents new structural insights and proposes novel models that challenge previous understandings.

---

## [Referee Report · Reviewer #3 (Public Review)]

Yin-wei Lin et al set out to visualize the inactive conformation of full length Bruton's Tyrosine Kinase (BTK), a molecule that has evaded high resolution structural studies in its full length form to this date. An open question in the field is how the Pleckstrin Homology-Tec Homology (PHTH) domain inhibits BTK activity, with multiple competing models in the field. The authors used a complimentary set of biophysical techniques combined with well thought out stabilizing mutations to obtain structural insights into BTK regulation in its full length form. They were able to crystallize the full length construct of BTK but unfortunately the PHTH was not resolved yielding the structure similar to previously obtained in the field. The investigation of the same construct by SAXS yielded an elongated structural model, consistent with previous SAXS studies. Using cryo-EM the authors obtained a low resolution model for the FL BTK with a loosely connected density assigned to the dynamic PHTH around the compact SH2-SH3-Kinase Domain (KD) core. To gain further molecular insights into PHTH-KD interactions the authors followed a previously reported strategy and generated a fusion of PHTH-KD with a longer linker, yielding a crystal structure with a novel PHTH-KD interface which they tested in biochemical assays. Lastly, Yin-wei Lin et al crystallized the BTK KD in a novel partially active state in a "face to face" dimer with kinases exchanging the activation loops, although partially disordered, being theoretically perfectly positioned for trans phosphorylation. Overall this presents a valiant effort to gain molecular insights into what clearly is a dynamic regulatory motif on BTK and is a valuable addition to the field.

I think the authors addressed all the comments that I had during the initial round of review. The only thing I can think of that would strengthen the paper is to add a supplemental figure/table with the results of unbiased SITUS fitting rather than just saying that it is close to manual fitting. Additionally, SITUS outputs not just one best solution but all the top fits and having a significant difference in cross correlation between the best fit and second best fit is usually indicative of true fit. As the authors already ran SITUS and colores they have this data and I think having a sup table with cross correlations for the top 3 fits for each of their maps would make their EM fitting more convincing and not hard to do.

Lastly, it seems like both the authors and I agree that the cryoEM reconstructions do not correspond to the reported resolutions by the FSC. This point in no way changes any of the conclusions of the paper, however, I can't help but feel guilty that some student who is not in the field will look at these EM maps in the future and think that this is how 7A reconstructions should look like. If the authors, maybe somewhere in the methods could add a sentence indicating that the FSC curves may be overly optimistic and that there are no secondary structure features present which would be expected at these resolutions, that would be great.

---

## [Author Response]

The following is the authors’ response to the original reviews.

**eLife assessment**
The study provides valuable insights into allosteric regulation of BTK, a non-receptor protein kinase, challenging previous models. Using a variety of biophysical and functional techniques, the paper presents evidence that the N-terminal PH-TH domain of BTK exists in a conformational ensemble surrounding a compact SH3-SH2-kinase core, that the BTK kinase domain can form partially active dimers, and that the PH domain can form a novel inhibitory interface after SH2/SH3 disengagement. Overall the presented evidence is solid, but the EM results may be over-interpreted and the work would benefit from additional functional validation.

We made every effort in our descriptions of the cryoEM data presented for full-length BTK to not overinterpret the results. In essence this is not an ideal EM target but given the failure by us and others to capture the full-length multi-domain protein crystallographically, we decided that the albeit low resolution cryoEM data are useful to the field.

**Reviewer #1 (Public Review):**
The manuscript by Lin et al describes a wide biophysical survey of the molecular mechanisms underlying full-length BTK regulation. This is a continuation of this lab's excellent work on deciphering the myriad levels of regulation of BTKs downstream of their activation by plasma membrane localised receptors.The manuscript uses a synergy of cryo EM, HDX-MS and mutational analysis to delve into the role of how the accessory domains modify the activity of the kinase domain. The manuscript essentially has three main novel insights into BTK regulation.1. Cryo EM and SAXS show that the PHTH region is dynamic compared to the conserved Src module.1. A 2nd generation tethered PH-kinase construct crystal of BTK reveals a unique orientation of the PH domain relative to the kinase domain, that is different from previous structures.1. A new structure of the kinase domain dimer shows how trans-phosphorylation can be achieved.Excitingly these structural works allow for the generation of a model of how BTK can act as a strict coincidence sensor for both activated BCR complex as well as PIP3 before it obtains full activity. To my eye the most exciting result of this work is describing how the PH domain can inhibit activity once the SH3/SH2 domain is disengaged, allowing for an additional level of regulatory control.I have very few experimental concerns as the methods and figures are well-described and clear. As the authors are potentially saying that the previously solved PH domain-kinase interface is artefactual, additional evidence strengthening their model would be helpful to resolve any possible controversies.

We do not argue that the previously solved PH domain-kinase interface is artefactual. Instead we point out that the PH/kinase interface identified in the prior structure is incompatible with the contacts between the SH3 and kinase domains in autoinhibited BTK. This then leads us to the suggestion that a PH/kinase inhibitory interaction may instead occur upon dissociation of the SH3-SH2 cassette from the kinase domain. Our data support that model. Moreover, our data suggest the PHTH domain is dynamic, likely not settling in to one particular autoinhibitory state. Thus, it is possible the previously solved PH/kinase structure exists within the conformational ensemble of a range PH/kinase domain interactions. In an effort to clarify our think we added two sentences to the Discussion (pg. 19).

**Reviewer #2 (Public Review):**
In this study, multiple biophysical techniques were employed to investigate the activation mechanism of BTK, a multi-domain non-receptor protein kinase. Previous studies have elucidated the inhibitory effects of the SH3 and SH2 domains on the kinase and the potential activation mechanism involving the membranebound PIP3 inducing transient dimerization of the PH-TH domain, which binds to lipids.The primary focus of the present study was on three new constructs: a full-length BTK construct, a construct where the PH-TH domain is connected to the kinase domain, and a construct featuring a kinase domain with a phosphomimetic at the autophosphorylation site Y551. The authors aimed to provide new insights into the autoinhibition and allosteric control of BTK.The study reports that SAXS analysis of the full-length BTK protein construct, along with cryoEM visualization of the PH-TH domain, supports a model in which the N-terminal PH-TH domain exists in a conformational ensemble surrounding a compact/autoinhibited SH3-SH2-kinase core. This finding is interesting because it contradicts previous models proposing that each globular domain is tightly packed within the core.Furthermore, the authors present a model for an inhibitory interaction between the N-lobe of the kinase and the PH-TH domain. This model is based on a study using a tethered complex with a longer tether than apreviously reported construct where the PH-TH domain was tightly attached to the kinase domain (ref 5). The authors argue that the new structure is relevant. However, this assertion requires further explanation and discussion, particularly considering that the functional assays used to assess the impact of mutating residues within the PH-TH/kinase domain contradict the results of the previous study (ref 5).

In our hands BTK activity is not significantly affected by mutation of just two residues, R133 and Y134. It is somewhat difficult to compare the previously reported activity assay for the same BTK mutant (Wang et al. ref 5, Figure 4D) with the data we report here. For unexplained reasons, the time scale for the quantitative assay in the previous work is truncated to 50 munutes for the R133/Y134 mutant data compared to 120 minutes for all of the other activity data reported in that figure. In our data, if we qualitatively examine the differences in a representative progress curve at 50 minutes between WT and the double R133/Y134 mutant (see Figure 6a, dark blue and pink traces) one might conclude that the R133/Y134 mutation is activating BTK. However, when we calculate the average kinase activity rate ± standard error for three independent experiments we find that the difference between WT and the double R133/Y134 mutant is not significant (see Figure 6b and c). Thus, instead of making any assertions about the previously published data we are trying to be as rigoruous as possible in presentation and interpretation of our own data.

In addition, throughout the manuscript we tried to be very careful in our discussion of our data and that published previously, to avoid conclusive statements about the previously described interface. Afterall, one of our overriding conclusions is that the N-terminal region of BTK is highly dynamic. See response to reviewer 1 above.

Additionally, the study presents the structure of the kinase domain with swapped activation loops in a dimeric form, representing a previously unseen structure along the trans-phosphorylation pathway. This structure holds potential relevance. To better understand its significance, employing a structure/function approach like the one described for the PH-TH/kinase domain interface would be beneficial.

We completely agree with this comment and are pursuing such studies now.

Overall, this study contributes to our understanding of the activation mechanism of BTK and sheds light on the autoinhibition and allosteric control of this protein kinase. It presents new structural insights and proposes novel models that challenge previous understandings. However, further investigation and discussion would significantly strengthen the study.

As indicated we are pursuing further investigation and felt that the body of work presented here is sufficient for a single manuscript.

**Reviewer #3 (Public Review):**
Yin-wei Lin et al set out to visualize the inactive conformation of full-length Bruton's Tyrosine Kinase (BTK), a molecule that has evaded high-resolution structural studies in its full-length form to this date. An open question in the field is how the Pleckstrin Homology-Tec Homology (PHTH) domain inhibits BTK activity, with multiple competing models in the field. The authors used a complimentary set of biophysical techniques combined with well-thought-out stabilizing mutations to obtain structural insights into BTK regulation in its full-length form. They were able to crystallize the full-length construct of BTK but unfortunately, the PHTH was not resolved yielding a structure similar to that previously obtained in the field. The investigation of the same construct by SAXS yielded an elongated structural model, consistent with previous SAXS studies. Using cryo-EM the authors obtained a low-resolution model for the FL BTK with a loosely connected density assigned to the dynamic PHTH around the compact SH2-SH3-Kinase Domain (KD) core. To gain further molecular insights into PHTH-KD interactions the authors followed a previously reported strategy and generated a fusion of PHTH-KD with a longer linker, yielding a crystal structure with a novel PHTH-KD interface which they tested in biochemical assays. Lastly, Yin-wei Lin et al crystallized the BTK KD in a novel partially active state in a "face-to-face" dimer with kinases exchanging the activation loops, although partially disordered, being theoretically perfectly positioned for transphosphorylation. Overall this presents a valiant effort to gain molecular insights into what clearly is a dynamic regulatory motif on BTK and is a valuable addition to the field.However, this work can be improved by considering these points:1. The cryo-EM reconstructions are potentially over-interpreted. The reported resolution for all of the analyzed reconstructions is better than 8Å, at which point helices should be recognized as well-resolved structural elements. In the current view/depiction of the cryo-EM maps/models it is hard to see such structural features and it would be great if the authors could include a panel showing maps at higher thresholds to show correspondence between the helices in the kinase C lobe and the cryo-EM maps. Otherwise, the overall positioning of the models within the cryo-EM maps is hard to evaluate and may very well be wrong. (Fig 4, S2).

First, we fully recognize the model is low-resolution and we are careful in our discussion of the cryo-EM data to use language that acknowledges the limitations of the model. Nevertheless, this is the model we have (specific data processing points are discussed below).

The resolution numbers are from the Fourier Shell Correlation (FSC) curve given by Cryosaprc at the end of refinement. We do acknowledge the reviewer’s comments that the resolution could be over estimated in that calculation, but our main focus is to show that the overall domain arrangement of the autoinhibited BTK core (Src-module) fits into the reconstructions.

We tested visualizing the maps at higher threshold, but the secondary structures of the reconstructions were still not well resolved. We do realize that with the current reconstructions, we do not have the structural details to correctly orientate and fit individual domains; this is why we chose to simply fit the available crystal structure of the autoinhibited BTK SH3-SH2-kinase core into the maps.

1. With the above in mind, if the maps are not at the point where helices are well resolved, it may be beneficial to low-pass filter the maps to a more conservative resolution for fitting, analysis, and representation. (Fig 4, S2).

Using low-pass filtered maps at 10Å or unsharpened maps, the fitting of the BTK model and map do not change significantly.

1. It would be valuable to get a quantitative metric on the model/map fitting for the cryo-EM work. One good package for this is Situs which provides cross-correlation values for the top orthogonal fits, without user input for initial fitting. This would again increase confidence in the correctness of model positioning on the map. (Fig 4, S2).

Thank you for this suggestion. We tested the colores feature (Exhaustive One-At-A-Time 6D Search) in Situs to perform model to map fitting without user input as the reviewer suggested. The highest ranked fitting is identical to what we presented in the manuscript. Following are the cross-corelation numbers calculated from “Fit-in-map” tool in chimera and from “collage” function in Situs. We now indicate this step in the caption to Figure 4.

**Author response table 1. sa4table1:** 

	Fit-in-map	Situs
Class 0	0.7181	0.6144
Class 1	0.7003	0.6277
Class 3	0.6273	0.5798

1. It would be great to see 2D class averages from the particles contributing to each of the 3D classes. Theoretically, a clear bright "blob" (hypothesized to be the PHTH domain) should be observable in the 2D class averages. In the current 2D class averages that region is unconvincingly weak. (Fig 4, S2).

We attempted to improve both 2D and 3D reconstructitions by feeding the particles from each 3D class through many cycles of 2D classification and selection to exclude ‘bad’ paritcles, but neither the 2D class averages nor 3D reconstructions could be improved.

We agree the feature that appears in the 2D class averages is weak. The BTK protein is only 77kD in size and is highly dynamic and flexible. Thus, in reality this is not an ideal system for cryo-EM. As well, the PHTH domain itself is quite small and NMR data, acquired in the context of a different project, provides evidence that the isolated PHTH domain is dynamic in solution (NMR linewidths vary throughout the protein suggesting intermediate exchange). Nevertheless, given the inability to capture the PHTH domain in crystal structures of full-llength BTK we reasoned that cryo-EM could provide some insight. In the future we anticipate building on these data to include inhibitory binding partners of BTK; however such an effort is beyond the scope of the current work.

1. It seems like there was quite a large circular mask applied during 2D classification. Are authors confident that the weak density attributed to the PHTH domain is not neighboring particles making their way into the extraction box? It would be great if the authors would trim their particle stack with a very stringent interparticle distance cutoff (or report the cutoff in the manuscript if already done so) to minimize this possibility.

We initially picked particles using a small radius (100 Å), and stringently selected 2D classes with particles that contained only density aligning to the core SH3-SH2-kinase domains. We found, however, that 3D ab initio reconstruction always resulted in an additional density located at different positions around the larger core density. The structure of a single BTK PHTH domain fits into that additional remote density. Given the additional density that consistently appeared in 3D reconstructions, we went back and picked particles using a larger circular mask (200 A). Subsequent 2D classification and 3D reconstruction from this analysis gave similar results and are presented in the manuscript.

Regardless of the mask radius, we used stringent conditions for particle picking and checked for the presence of duplicates. An interparticle distance cutoff of 0.1 to 0.5 times the particle diameter was used and resulted in fewer number of particles, but the presence of the extended density remains. We also made use of template picking (2D class averages) to repick the particles and found no significant difference in the number of particles or quality of 2D classifications.

1. The cryo-EM processing may benefit from more stringent particle picking. The authors picked over 2M particles from 750 micrographs which likely represents very heavy overpicking. I would encourage the authors to re-pick the micrographs with 2D class averages and use more stringent metrics to reduce the overpicking. This may result in higher-resolution reconstructions. (Fig 4, S2).

This was an effort to maximize the number of particles extracted. After multiple rounds of 2D classification and selection to exclude empty and junk particles, the final number of particles selected for 3D ab-initio reconstructions were only 68,788, and only ~20K particles for each 3D reconstruction. Thus, we are not concerned that we overpicked particles. This approach is described in Supp Figure S2.

1. The Dmax from SAXS for the Full Length BTK is at 190Å. It would be great if the authors could make a cartoon of what domain arrangement may satisfy this distance, as it is quite extended for such a small particle. Can the authors rule out dimerization at SAXS concentrations? (Fig 1).

SAXS data for full-length, wild-type BTK has been previously published (Márquez et al, 2003 EMBO J. (2003) 22:4616-4624). Our data for WT BTK are consistent with that published previously (and we have cited this previous work). In that work, the authors attribute the ~200 Å Dmax value to an elongated BTK conformation where the domains of BTK are arranged in a linear fashion (a figure showing this domain arragement is provided by Marquez et al. precluding the need for such a cartoon here).

In the present work we take advantage of targeted mutations to stabilize the autoinhibted SH2-SH2-kinase core and the Dmax value that we report for this more autoinhibited version of full-length BTK (FL 4P1F) is ~150Å. Notwithstanding low resolution in both SAXS and cryoEM, it is notable that superposition of the cryoEM models in Figure 4c & d gives a distance of ~150Å between the PHTH domains from the two models.

Finally, we cannot completely rule out that a small fraction of full length BTK is forming dimers. However, in our experience purifying and working with this protein, we find that purified and concentrated monomeric fulllength Btk proteins (as high as 15mg/ml) are quite stable and remain monomeric and free of aggregation even after sitting at 4°C for more than a week. Here the BTK SAXS data were collected within 24 hours after the samples were thawed.

1. In Figure S1 (C) it seems that the curves are just scattering curves with Guinier plots in the inserts, but are labeled as Guinier plots in the legend. The Guinier plots for some samples (FL 4P1F) show signs of aggregation, which may complicate the analysis, it could be beneficial to redo.

We thank the reviewer for pointing out our mistake in presention of the SAXS data. We have now replaced plots in Figure S1c with the correct scattering profiles for each construct with the Guinier insets shown. We revised the label of this panel to “Scattering profile and Guinier plots (insets)”.

In addition, we re-processed the FL 4P1F data by performing buffer subtraction (using a different buffer alone scattering dataset (also collected during original data acquisition)). The data quality after reprocessing were significantly improved (see new scattering profiles and Guinier plots for full-length BTK inSupplementary Figure S1). Protein stability (see above) and the current data quality therefore suggest that aggregation is not complicating the SAXS analysis.

1. Have the authors verified that the activation loop mutations that they introduce do not disrupt the PHTH binding as they previously reported an activation loop on BTK to interact with PHTH, an interaction they do not see here? If so, a citation would be helpful in the text. If not, testing this would strengthen the paper.

The same activation loop mutations were included in the constructs used in the previous solution studies of the PHTH/kinase domain interaction by NMR and HDX (see ref [11]). We clarify this point in the methods section. As well, all but one of the sequence changes introduced into the activation loop are at positions at the ‘base’ of the activation loop and therefore are not surface exposed. Only one amino acid change is on the exposed part of the activation loop (V555T).

1. Can the authors comment on the surfaces which are accessible and inaccessible to the PHTH in the crystal (Fig 3E)? The fact that PHTH doesn't adopt a stable conformation in the solvent channel to some degree indicates that the accessible interaction surfaces are not suitable for PHTH interactions, as the "effective concentration" of the PHTH would be quite high. Are these surfaces consistent with the cryo-EM analysis?

This is an excellent point and we did state the following in describing the crystallization results:

“the crystallography results are consistent with a flexible N-terminal PHTH domain with the caveat that the domain swapped dimer organization might limit native autoinhibitory contacts between the PHTH and SH3SH2-kinase regions.”

In the domain swapped dimer seen in the crystal, a symmetry related molecule does partially block the Ghelix region of the kinase domain while the activation loop and C-helix in the N-lobe remain accessible. Our previous solution studies (ref [11]) pointed to the G helix as part of the interaction interface in addition to the activation loop and part of the N-lobe. We have now modified the sentence above to more clearly describe which parts of the kinase domain are inaccessible in the crystal and the possible ramifications of the steric environment on PHTH domain mobility in the crystal (see pg. 10). That said, all of our previous HDX data shows little protection in the PHTH domain in full-length BTK (mapping of the PHTH/kinase interaction was only possible in trans using excess PHTH domain) and so our data can be best summarized by concluding that the PHTH domain visits a number of conformational states and makes transient contacts with various regions of the kinase domain (dependent upon whether the SH3-SH2 region is engaged or not). This is similar to the ‘fuzzy’ intramolecular contacts described for the N-terminal region of the SRC family. Like the SRC family, BTK (and other TEC kinases) contain a long disordered linker between the N-terminal region and the compact SH3-SH2-kinase core.

1. For the novel active state dimer of the Kinase Domain it would be great to see some functional validation of the dimerization interface. It is structurally certainly quite suggestive, but without such experiments the functional significance is unclear. If appropriate mutations have been published previously a citation would be helpful.

We completely agree. We scoured the literature and our own facuntional assay results over many years but the appropriate mutations to test the functional significance of the kinase domain dimer have not been reported or previously studied in our lab. We are therefore actively pursuing this line of investigation now.

**Reviewer #1 (Recommendations For The Authors):**
I have the following proposed experiments/analysis that should help.1. To better validate the putative PH-kinase interface seen, the authors should try some alphafold multimer / rosettaTTFold modelling of just the PHTH module with the kinase domain. The advantage of this is that it will test how conserved over evolution the potential interface is, and will help to decipher discrepancies between the two structures. This may end up being similar to what is seen in Akt (in this case the alphafold prediction does not match the allosteric inhibitor structure, or the nanobody bound structure), but this could help provide additional insight into how the PH domain interacts.

We have applied alphafold to this system. The PHTH-kinase fusion sequence was fed to Alphafold and the separate PHTH and kinase domains to Aphafold multimer. The results provide a range of ‘complexes’ none of which recapitulate the PHTH/kinase interface reported here or that reported by Wang et al in previous work. Three of five results from Alphafold Multimer place the PHTH domain on the activation loop face of the kinase domain consistent with the previous solution data pointing to a similar regulatory interface. This is interesting but our experience in applying alphafold to dynamic confromationally heterogeneous systems is that the results need to be considered with caution. For that reason we did not include any of the alphafold predictions in the manuscript.

Evolutionary conservation is discussed further in the next section:

1. Could the authors provide a detailed evolutionarily analysis of the binding surface between the PHTH and kinase domains and include this in Fig5, this also would help interpret the likelihood of this interface.

This is an excellent question and we have in fact previously published a detailed evolutionary analysis of the BTK kinase domain in collaboration with Kannan Natarajan (see Amatya et al., PNAS, 2019, [ref 11]). In that work we found that evolutionarily conserved residues on the kinase domain map to the activation loop face, supporting the solution data that the PHTH interacts with the kinase domain across the activation loop face. That work predated alphafold but it is interesting that, to the exent that alphafold predicts anything, it seems to converge on the PHTH domain containg the activation loop face.

In the context of our current work, and this question from the reviewer, we re-examined the evolutionary anlysis carried out previously and find that BTK (or TEC family) specific residues on the kinase domain do not appear at the newly identified PHTH/kinase interface we report here. We could speculate that since the ‘back’ of the kinase domain N-lobe interacts with multiple binding partners (SH3, SH2-linker and PHTH) evolutionary pressures may have resulted in a certain degree of plasticity to allow recognition of multiple binding partners.

Evolutionary analysis of the BTK PH domain was also carried out previously and shows that the conserved sites map to the phospholipid binding pocket of the PH domain. The analysis did not include TH domain residues. Since we find the TH domain contributes to the PHTH/kinase interface in our crystal structure, we do not have the data at this time to do a thourough anaylsis but we appreciate this comment and can address this in furture work with collaborators.